

# Numerical reconstructions of the flow and basal conditions of the Rhine glacier, European Central Alps, at the Last Glacial Maximum

Denis Cohen[1], Fabien Gillet-Chaulet[2], Wilfried Haeberli[3], Horst Machguth[3,4], and Urs H. Fischer[5]

[1]Department of Earth and Environmental Science, New Mexico Tech, Socorro, NM, USA
[2]Univ. Grenoble Alpes, CNRS, IRD, Grenoble INP, IGE, Grenoble, France
[3]Department of Geography, University of Zurich, Zurich, Switzerland
[4]Department of Geosciences, University of Fribourg, Fribourg, Switzerland
[5]Nagra, Wettingen, Switzerland

*Correspondence to:* Denis Cohen (denis.cohen@gmail.com)

**Abstract.** At the Last Glacial Maximum (LGM), the Rhine glacier in the Swiss Alps covered an area of about 16,000 km$^2$. As part of an integrative study about the safety of repositories for radioactive waste under ice age conditions in Switzerland, we modeled the Rhine glacier using a fully-coupled, three-dimensional, transient, thermo-mechanical Stokes flow model down to a horizontal resolution of about 500 m. The accumulation and ablation gradients that roughly reproduced the geomorphic

reconstructions of glacial extent and ice thickness suggested extremely cold ($T_{\text{July}} \sim 0°C$ at the glacier terminus) and dry ($\sim 10$ to 20% of today's precipitation) climatic conditions. Forcing the numerical simulations with warmer and wetter conditions that better matched LGM climate proxy records yielded a glacier on average 500 to 700 m thicker than geomorphic reconstructions. Mass balance gradients also controlled ice velocities, fluxes, and sliding speeds. These gradients, however, had only a small effect on basal conditions. All simulations indicated that basal ice reached the pressure melting point over much of the Rhine

and Linth piedmont lobes, and also in the glacial valleys that fed these lobes. Only the outer margin of the lobes, bedrock highs beneath the lobes, and Alpine valleys at high elevations in the accumulation zone remained cold based. The Rhine glacier was thus polythermal. Sliding speed estimated with a linear sliding rule ranged from 20 to 100 m a$^{-1}$ in the lobes, and 50 to 250 m a$^{-1}$ in Alpine valleys. Velocity ratios (sliding to surface speeds) were $> 80\%$ (lobes) and $\sim 60\%$ (valleys). Basal shear stress was very low in the lobes (0.03–0.1 MPa), much higher in Alpine valleys ($> 0.2$ MPa). In these valleys, viscous strain heating

was a dominant source of heat, particularly when shear rates in the ice increased due to flow constrictions, confluences, or flow past large bedrock obstacles, contributing locally up to several W m$^{-2}$ but on average 0.03 to 0.2 W m$^{-2}$. Basal friction acted as a heat source at the bed of about 0.02 W m$^{-2}$, 4 to 6 times less than the geothermal heat flow which is locally high (up to 0.12 W m$^{-2}$). In the lobes, despite low surface slopes and low basal shear stresses, sliding dictated main fluxes of ice which closely followed bedrock topography: ice was channeled in between bedrock highs along troughs, some of which coincided

with glacially eroded overdeepenings. These sliding conditions may have favored glacial erosion by abrasion and quarrying. Our results confirmed general earlier findings but provided more insights into the detailed flow and basal conditions of the Rhine glacier at the LGM. Our model results suggested that the trimline could have been buried by a significant thickness of cold ice. These findings have significant implications for interpreting trimlines in the Alps and for our understanding of ice-climate interactions.





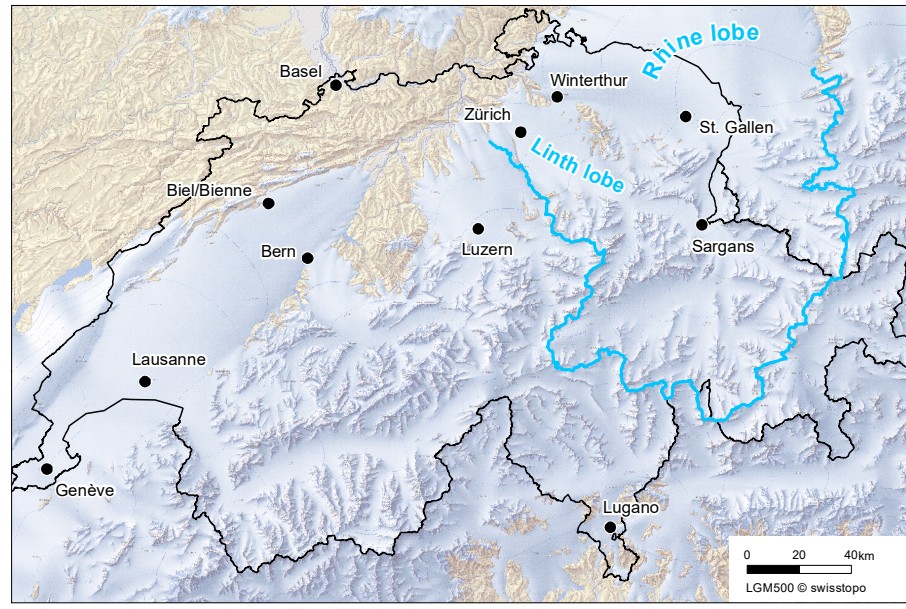

**Figure 1.** Map of Swiss Alps at the LGM showing maximum extent of ice cover (from Bini et al., 2009) with outline of Rhine glacier basin in blue. Source: Bundesamt für Landestopografie swisstopo.

# 1 Introduction

During the Last Glacial Maximum (LGM), the Alps were heavily glaciated and the Rhine glacier formed a large transection glacial complex that drained ice from the Alps north into the central Swiss and southwestern German Alpine Forelands (Fig. 1). Repeated glacial advances into the lowlands during the Pleistocene (Preusser et al., 2011) sculpted the present-day landscape

forming emblematic valleys, horns, and arêtes in the Alps, deep, narrow lakes covering glacially-excavated overdeepenings partially filled with glacial deposits, and moraines, outwash planes, terraces, and other depositional landforms in the lowlands. Numerous geomorphic studies since the beginning of the twentieth century have helped constrain the geometry and some flow characteristics of the Rhine glacier at its maximum extent during the last glaciation. Geomorphic mappings of terminal and lateral moraines delineated the extent of ice advances at glacial maxima and at various intermediate positions (e.g., Penck and

Brückner, 1909; Schlüchter, 1988; Keller, 1988; Schlüchter, 2004; Beckenbach et al., 2014). Erratics, tills and other sediment deposits provided information about flow paths and provenances (e.g., Haeberli and Schlüchter, 1987; Florineth, 1998; Anselmetti et al., 2010; Ellwanger et al., 2011; Braakhekke et al., 2016). Presence of till indicated depositional environments while evidences of quarried bedrock, glacial polish, and striated rock surfaces identified areas with predominantly temperate (at the pressure melting point of ice), wet-based, erosive basal conditions (Florineth, 1998; Beckenbach et al., 2014). In the accumu-

lation area, trimlines represent the upper limit of glacial erosion and define the minimum elevation of the ice surface (Florineth and Schlüchter, 1998; Kelly et al., 2004). This large amount of geomorphic information for the Rhine glacier, probably the best documented of any paleo glacier, has been used to create detailed maps of reconstructed ice extent and ice surface elevation at



the LGM (e.g., Jäckli, 1962; Jäckli, 1970; Keller and Krayss, 1987, 1989, 1993, 1994; Benz-Meier, 2003; Kelly et al., 2004; Bini et al., 2009).

These detailed maps have been used to infer quantitative glaciological characteristics of the Rhine glacier at the LGM. In the ablation area, the first paleo-glaciological studies using simplified two-dimensional models (Blatter and Haeberli, 1984;
Haeberli and Penz, 1985) indicated that thin, flat and extended lobes of the large piedmont glaciers spreading out over much of the Swiss Plateau were polythermal with temperate basal conditions and low driving stresses (about 0.03 MPa, Haeberli and Penz, 1985). At the LGM, central Europe experienced extremely cold and dry conditions with the penetration of winter sea ice to low latitudes in the Atlantic Ocean. Correspondingly, the closure of the primary humidity source north of the Alps (Florineth, 1998; Florineth and Schlüchter, 2000; Hofer et al., 2012) implied that most of the moisture feeding glaciers in the
Alps had a southern, Mediterranean origin (Luetscher et al., 2015). As a result, glaciers north of the ice divide had small ice throughflow with small surface velocities. In the northern lobes of piedmont glaciers, estimated average surface velocity was only about 25 m a$^{-1}$ for the Rhine lobe (Haeberli and Penz, 1985). Average ice flow velocity through the main outlets of the Rhine glacier was estimated to be less than 200 m a$^{-1}$ (Keller and Krayss, 2005b), a value relatively small compared to present-day ice bodies of comparable size. Extreme cold conditions also favored the development of permafrost up to about
150 m thick north of the marginal zone of the Rhine glacier (Haeberli et al., 1984; Delisle et al., 2003; Lindgren et al., 2016). Subsurface temperatures and groundwater flow conditions must have been strongly influenced by the presence of extended surface and subsurface ice (Speck, 1994). Under these conditions, the glacier margins likely consisted of cold ice frozen to the subglacial permafrost, with limited basal sliding and basal melt-water flow, conditions that would have prevented significant glacial erosion there. Yet, erosional features such as overdeepenings are found close to earlier positions of the ice margin of
the Rhine glacier (Fig. 2; Preusser et al., 2011; Dehnert et al., 2012), necessitating basal conditions favorable for erosion. These could have occurred as a result of more humid conditions with higher ice flow velocity and increased sliding during ice advances across the Swiss Plateau, or during the rapid down-wasting of the ice mass that is likely to have taken place during the retreat phases, generating large quantities of water necessary for subglacial erosion. Another possibility is excavation during earlier ice ages with larger ice extents and warm-based conditions at sites with cold and frozen ice margins during the
LGM. Alternatively, these early simple models may not have captured the full complexity of basal conditions near the glacier terminus at the LGM. Early two-dimensional models of the Rhine glacier at the LGM (Blatter and Haeberli, 1984; Haeberli and Penz, 1985) were not able to include the complexity of ice flow inherent to transection glaciers. Simple one-dimensional gravity-driven flow approximations of flow velocities from three-dimensional ice surface reconstructions (e.g., Benz-Meier, 2003) provided interesting first-order results but are physically not consistent when sliding is an important component of flow,
neglected important stresses (longitudinal and transverse) relevant to the complex ice flow patterns of transection glaciers (Kirchner et al., 2016), and ignored glacier thermodynamics. An estimate of potential erosion for the Rhine glacier using this approximation (Dürst Stucki and Schlunegger, 2013) did not find a correlation between ice sliding speed and areas of overdeepenings in the Rhine lobe.

Because of uncertainties in the reconstructed ice surface geometry and derived glaciological quantities in both the accumu-
lation zone (more limited field evidences, trimline uncertainties, poorly known accumulation at the LGM) and in the ablation





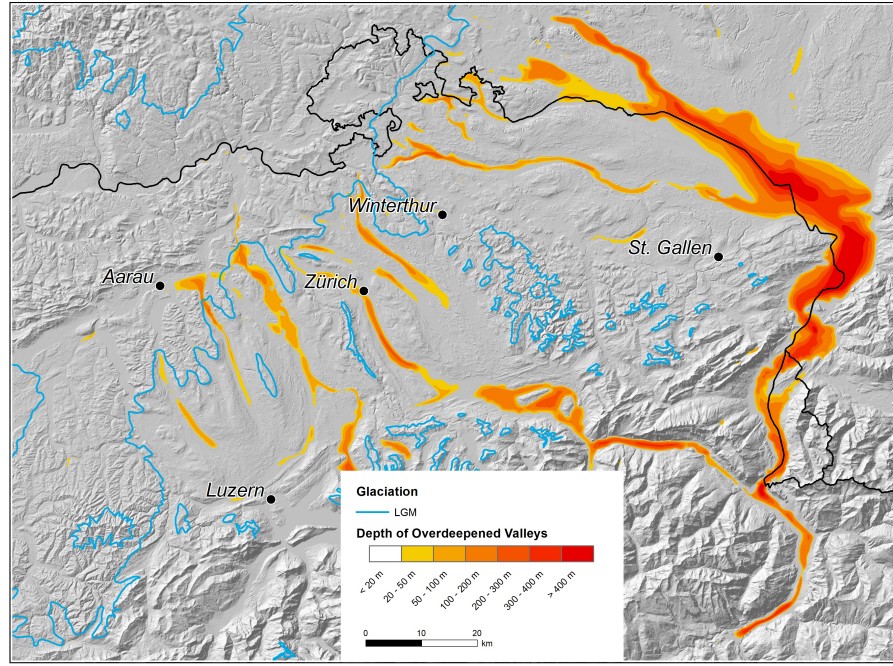

**Figure 2.** Depth of overdeepened valleys in the Rhine glacier basin (Pietsch and Jordan, 2014). Outline of maximum glacial extent at LGM (Bini et al., 2009) is shown in blue.

zone (poorly constrained temperate versus frozen basal conditions, ice flux, basal shear stress, and melt rates), geomorphic reconstructions of ice surface geometry should be verified against a three-dimensional ice flow model of the Rhine glacier. Flow conditions (patterns, velocities, stresses) obtained from geomorphic ice surface reconstruction (e.g., Benz-Meier, 2003; Dürst Stucki and Schlunegger, 2013) should abide by the fundamental principles of glacier dynamics (Tarasov et al., 2012; Stokes et al., 2015). For example surface flow directions should follow major ice drainages and computed vertical surface velocity from reconstructed ice surface should match mass balance conditions imposed by the LGM climate (e.g., Haeberli, 1991). These conditions may not always be satisfied with geomorphically reconstructed ice surfaces and warrant alternative approaches for paleo-reconstructions of ice bodies. The limits of geomorphic reconstructions and of simple stress-driven calculations necessitate a numerical modeling approach that can estimate more precisely the flow of the Rhine glacier at the LGM.

Finally, knowing the areas of temperate basal conditions, where sliding dominates, is important for addressing the safety of deep geological repositories for radioactive wastes. The long-term management of these wastes produced through the use of radioactive materials in power production, industry, research, and medicine entails their containment and isolation for over hundreds of thousands of years. Over such extended time periods, the performance of repositories in mid- and high-latitude regions can be affected by impacts from future ice-age conditions. Several countries have developed programs to investigate potential future ice-related environmental changes and their effects at depth (Fischer et al., 2015). The main concerns are deep





erosion by glaciers or ice sheets, penetration of permafrost to great depth, changes in groundwater hydrology due to permafrost and ice loading and their complex interactions. To address these issues, two main sources of information are being used: (i) qualitative and quantitative information about regional climate and ice conditions during past ice ages, and especially the LGM, as interpreted from paleo-climatic and paleo-ice proxies, and (ii) numerical modeling of complex and strongly coupled

ice/climate systems. The present study was carried out within the framework of considerations concerning the long-term safety of repositories for radioactive waste in northern Switzerland (see Fig. 3).

Here, we use a fully three-dimensional, numerical, thermo-mechanical ice flow model that solves Stokes equations to investigate in detail the general characteristics of the Rhine glacier and to critically reflect on the accuracy of the geomorphic reconstructions with respect to ice flow physics and LGM climate. The computational burden of solving Stokes equations

implies that short (a few thousand years at most) transient simulations around the LGM are used only to seek steady state solutions, neglecting transient effects of climate and other processes (e.g., isostatic adjustment). Stokes equations are solved using Elmer/Ice (Gagliardini et al., 2013), an open source finite element code for ice flow. The ice-flow model is driven by a simple mass balance model parameterized by two mass balance gradients, one for the accumulation zone, one for the ablation zone, and a given equilibrium line altitude (ELA). Parameterization of temperature is based on a given temperature at the ELA and a

lapse rate. The model yields the full three-dimensional velocity and temperature fields, details of surface, englacial, and sliding speeds, basal temperatures and shear stresses from which ice fluxes, flow patterns, and areas of temperate basal condition can be derived.

## 2   The Rhine glacier model

We model the flow of the Rhine glacier using the full equations of mass and momentum, the Stokes flow equations, coupled

to the heat equation over the Rhine glacier basin (Fig. 1) at a horizontal resolution of about 500 m using the finite element, open source code Elmer/Ice (Gagliardini et al., 2013). Despite its significant computational cost, Stokes flow is more accurate than shallow ice/shallow shelf approximation in zones where gradients in stresses are significant like in complex transection glaciers found in Alpine settings and were sliding is also significant (Ryser et al., 2014; Kirchner et al., 2016). Because of computational cost, transient simulations are limited to several thousand years at most, seeking steady state solutions for

a constant LGM climate (temperature, mass balance gradients). Our domain of computation for the Rhine glacier includes all basins that drain into present-day Lake Constance and Lake Zurich. These basins straddle four countries: Switzerland, Germany, Liechtenstein, and Austria. Present-day topographic divides with the Rhone basin to the west, the Ticino basin to the south, and the Inn basin to the south-east delineate sources of ice in the accumulation area. For the ablation area to the north, the maximum glacial extent at the LGM is extended about 50 km northward of the Rhine and Linth lobes to delimit the

extent of the model. This allows the Rhine glacier to advance further north than its LGM extent in the numerical simulations if necessary. On the western side, the divide between the Reuss and Linth lobes serves as the model boundary. The eastern limit of the Rhine lobe is used to delineate the eastern edge of the model.





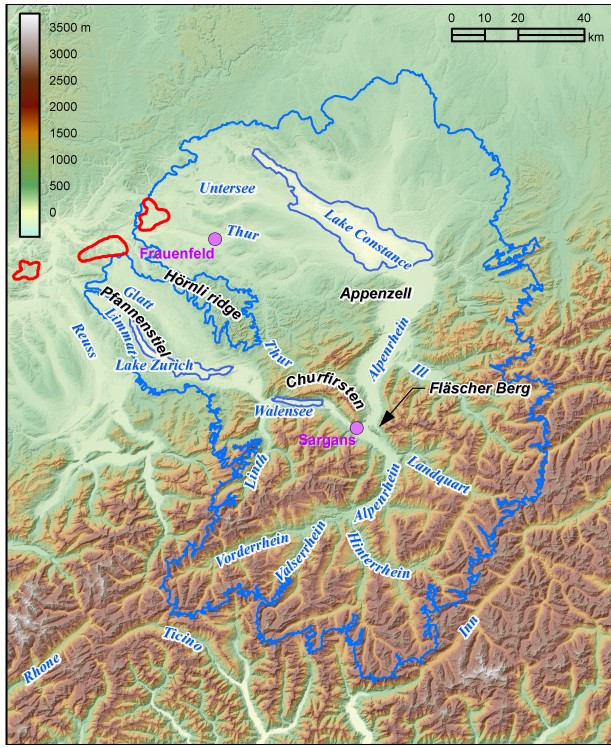

**Figure 3.** Basal topography (from Benz-Meier, 2003) used in all simulations with names of major valleys, present-day lakes, cities, and mountains mentioned in text. The blue outline of the Rhine glacier at the LGM is from Benz-Meier (2003). See text for details. The three proposed siting regions for high-level waste repositories in Switzerland are shown in red. All three sites are located within the ice extent of earlier glaciations larger than the LGM.

The basal topography of our ice flow model uses the present-day topography (initially obtained from a 25 m resolution digital elevation model). This topography was modified to remove present-day ice thickness using published results of an inversion model by Linsbauer and colleagues (Linsbauer et al., 2012; Paul and Linsbauer, 2012), lake bathymetry where available (Lake Constance and Lake Zurich, Benz-Meier, 2003), and by depressing the topography to reproduce isostatic adjustments at the

5   LGM (up to 130 m) using the model of Norton and Hampel (2010). Figure 3 shows the basal map used for the model. For the ice surface, we use either the digitally available reconstructed glacier surface of Benz-Meier (2003) or an ice surface obtained from a previous numerical simulation. The reconstruction of Benz-Meier (2003) is based on the earlier works of Jäckli (Jäckli, 1962; Jäckli, 1970) and also of Keller and Krayss (Keller and Krayss, 1987, 1989, 1993, 1994). We refer to it as Benz-Meier (2003) despite its earlier origin. A two-dimensional, fixed, unstructured triangular mesh over the domain is created using Gmsh

10  (Geuzaine and Remacle, 2009) and then extruded into hexahedral elements in the vertical direction between the basal surface and the ice surface.

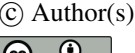



## 2.1 Thermomechanical model

Stokes equations describe the mass and momentum balance for a viscous fluid. Together with the equation for energy, this system forms a thermo-mechanical problem of five coupled scalar equations with five field variables to solve for: $p$, the pressure, $\boldsymbol{v}$ the velocity vector ($u$, $v$, $w$), and $T$, the temperature. In addition, for ice flow, the glacier surface elevation, $\zeta$, is unknown and requires an additional equation, the kinematic boundary condition.

## 2.2 Stokes equations

The conservation of mass for an incompressible fluid (divergence-free velocity field) is

$$\nabla \cdot \boldsymbol{v} = 0, \tag{1}$$

where $\nabla \cdot$ indicates the divergence. The conservation of momentum is

$$-\nabla p + \eta \nabla^2 \boldsymbol{v} + \left[\nabla \boldsymbol{v} + (\nabla \boldsymbol{v})^{\mathrm{T}}\right] \nabla \eta + \rho \boldsymbol{g} = 0, \tag{2}$$

where $\boldsymbol{g}$ is the vector of gravity, $\rho$ is the density of ice, $\nabla$ indicates the gradient, and $\eta$ is the ice viscosity given by Glen's flow law

$$\eta = \begin{cases} \frac{1}{2} \left(EA(T)\right)^{-1/n} d^{-(1-1/n)}, d > d_o, \\ \frac{1}{2} \left(EA(T)\right)^{-1/n} d_o^{-(1-1/n)}, d \leq d_o, \end{cases} \tag{3}$$

where $d = \sqrt{\left(\frac{1}{2}\mathrm{tr}(D^2)\right)}$ is the effective strain rate, $d_o$ is the critical strain rate, $D = \frac{1}{2}\left[(\nabla \boldsymbol{v}) + (\nabla \boldsymbol{v})^{\mathrm{T}}\right]$ the strain-rate tensor, $n$ the power-law exponent, $T$ the temperature, $A(T)$ the rate factor, and $E$ the flow-enhancement factor taken equal to 1 in all simulations. An Arrhenius relationship is used for $A$,

$$A(T) = A_o(T) \exp\left(\frac{-Q(T)}{RT}\right) \tag{4}$$

where $A_o$ is the pre-factor, $Q$ is the activation energy, and $R$ is the gas constant. Different values of $A_o$ and $Q$ are used depending on whether $T$ is greater or less than $-10°\mathrm{C}$.

## 2.3 The heat equation

The heat equation for a viscous fluid is

$$\rho c(T) \left(\frac{\partial T}{\partial t} + \boldsymbol{v} \nabla T\right) = \nabla \left(\kappa(T) \nabla T\right) + 4\eta d^2, \tag{5}$$

where $c(T)$ is the heat capacity of ice and $\kappa(T)$ the heat diffusivity of ice, both functions of temperature. Parameterization of these equations and values of model parameters for the ice flow model are given in Table 1. The last term in the equation above describes viscous dissipation.



**Table 1.** Model parameter values.

| Parameter | Value | Units | Description |
|---|---|---|---|
| $\rho$ | 917 | $\mathrm{kg\,m^3}$ | Density of ice |
| $\boldsymbol{g}$ | $(0,0,-9.81)$ | $\mathrm{m\,s^{-2}}$ | Gravitational acceleration |
| $n$ | 3 | | Glen exponent |
| $A_o$ | $1.258 \times 10^{13}$ | $\mathrm{MPa^3\,a^{-1}}$ | Pre-factor for $T < -10^\circ\mathrm{C}$ |
| | $6.046 \times 10^{28}$ | $\mathrm{MPa^3\,a^{-1}}$ | Pre-factor for $T \geq -10^\circ\mathrm{C}$ |
| $Q$ | $60.0 \times 10^3$ | $\mathrm{J\,mol^{-1}}$ | Activation energy for $T < -10^\circ\mathrm{C}$ |
| | $139.0 \times 10^3$ | $\mathrm{J\,mol^{-1}}$ | Activation energy for $T \geq -10^\circ\mathrm{C}$ |
| $R$ | 8.314 | $\mathrm{J\,mol^{-1}\,K^{-1}}$ | Gas constant |
| $d_o$ | $3.16 \times 10^{-2}$ | $\mathrm{a^{-1}}$ | Critical strain rate |
| $k$ | $9.828\exp\left(-5.7 \times 10^3\,T\right)$ | $\mathrm{Wm^{-1}K^{-1}}$ | Heat conductivity of ice. $T$ in K |
| $c$ | $146.3 + 7.253\,T$ | $\mathrm{J\,K^{-1}}$ | Heat capacity of ice. $T$ in K |
| $T_{\mathrm{pmp}}$ | $273.15 - 9.8 \times 10^{-8}\,p$ | K | Pressure melting point. $p$ is pressure in Pa |
| $m$ | 1 | | Sliding exponent |
| $C_o$ | 0.001 | $\mathrm{MPa\,a\,m^{-1}}$ | Sliding parameter for temperate ice |
| $C_1$ | 0.1 | $\mathrm{MPa\,a\,m^{-1}}$ | Sliding parameter for cold ice |
| $\gamma$ | 2 | K | Sub-melt sliding parameter |

## 2.4 Basal boundary conditions

### 2.4.1 Geothermal heat flow

Geothermal heat flow in the area of the LGM Rhine glacier is highly variable ranging from 0.06 to 0.12 W m$^{-2}$ with low values in the high Alps and high values in geothermally active regions of the Swiss Plateau. We use the heat flow data of Medici and Rybach (1995) obtained from numerous measurements of temperature gradients in boreholes and of rock thermal properties. Present observed temperature gradients, however, are affected by topographic effects, long-term changes in climate and erosion, groundwater flow, and past ice sheet cover in a complex, non-linear way. Correcting for these effects is difficult and is not done in Medici and Rybach (1995). Neither is it done here. It would require, at a minimum, to impose the heat flux at the base of a 2 to 3 km thick bedrock below the ice and hence would necessitate solving the heat equation in the bedrock beneath the ice, a task beyond the scope of the present study.

### 2.4.2 Basal sliding

At the bed, ice can slide over bedrock when water is present which occurs when the temperature at the bed equals the melting temperature (so called temperate bed). Sub-freezing sliding can also occur (e.g., Shreve, 1984) but is small and negligible for





ice fluxes or erosion for the time scale considered here (100s to 1000s years). It is neglected here. A commonly-used approach for the sliding speed is to link it to the basal shear stress, i.e.,

$$\boldsymbol{\tau}_b = C \|\boldsymbol{v}_s\|^{m-1} \boldsymbol{v}_s, \tag{6}$$

where $\boldsymbol{v}_s$ is the sliding speed vector, $\boldsymbol{\tau}_b$ the basal shear stress vector, $m$ is an exponent, and $C$ is a constant that encapsulates

the effects of water pressure, bed roughness, etc. The parameter $m$ usually takes values between 1 and $1/3$. In all simulations we assume $m = 1$ (linear sliding).

Ice sliding over the substrate occurs when temperatures reach the melting temperature. Below it $\boldsymbol{v}_s = \boldsymbol{0}$. To simulate this behavior we make the sliding coefficient $C$ in the sliding rule above a function of temperature $T$ (Greve and Blatter, 2009),

$$C(T) = (C_o - C_1) \exp\left[-\frac{T'}{\gamma}\right] + C_1 \tag{7}$$

where $C_o$ and $C_1$ are the sliding coefficients for temperate and cold conditions, respectively, and $T' = T - T_{\mathrm{pmp}}$ where $T_{\mathrm{pmp}}$ is the temperature at the pressure melting point. The parameter $\gamma$, sometimes called the sub-melt sliding parameter (Seddik et al., 2012), adjusts the range of temperature over which this transition occurs. For numerical stability we use $\gamma = 2$.

### 2.4.3 Thermal boundary condition

The boundary condition for the basal temperature is (Seddik et al., 2012):

$$\kappa(T)\nabla T \cdot \boldsymbol{n} = q_{\mathrm{geo}} - \boldsymbol{t} \cdot \boldsymbol{v}_s, \tag{8}$$

where $\boldsymbol{n}$ is the normal to the glacier bed, $q_{\mathrm{geo}}$ is the geothermal heat flux, and $\boldsymbol{t}$ is the stress vector at the glacier bed. The last term of the equation is the thermal energy due to basal friction. In most simulations this term is assumed to be zero. When the glacier bed is at the pressure melting point, Eq. (8) no longer holds but the difference between the left and right sides of this equation can be used to compute the melt rate.

## 2.5 Boundary conditions at the ice surface

Solving for ice velocity, temperature, and the elevation of the ice surface requires three types of boundary condition at the ice surface: (i) a kinematic boundary condition that describes the motion of the ice surface as a function of accumulation (or ablation) and ice flow, (ii) an expression of the stress-free condition at the ice surface, (iii) and the ice temperature.

### 2.5.1 Stress-free surface

The glacier surface is a free surface in contact with the atmosphere. This surface supports no shear stress. This is simply expressed as $\boldsymbol{t} \cdot \boldsymbol{n} = 0$, where $\boldsymbol{t}$ is the stress vector at the ice surface and $\boldsymbol{n}$ is the unit normal pointing outward.



### 2.5.2 Kinematic boundary condition

The ice surface moves up or down depending on the balance between mass flux across the glacier surface and the divergence of the velocity field. This is expressed mathematically by the kinematic boundary condition

$$\frac{\partial \zeta}{\partial t} = \dot{b} + w - u\frac{\partial \zeta}{\partial x} - v\frac{\partial \zeta}{\partial y}, \tag{9}$$

where $\zeta$ is the glacier surface elevation and $\dot{b}$ is the specific balance rate (specific balance for short), i.e., the volumetric mass
flux of ice per unit time across the glacier surface, the accumulation/ablation function.

### 2.5.3 Surface mass balance

For the accumulation/ablation function ($\dot{b}$ in Eq. (9)), we choose a simple parameterization represented by two mass balance gradients, one for the accumulation area, one for the ablation area, and a maximum threshold value for the maximum accumulation,

$$\dot{b} = \begin{cases} \min\left(\dot{b}_{\mathrm{acc}}^{\max}, \beta_{\mathrm{acc}}(z - z_{\mathrm{ela}})\right), & z \geq z_{\mathrm{ela}} \\ \beta_{\mathrm{abl}}(z - z_{\mathrm{ela}}), & z < z_{\mathrm{ela}}, \end{cases} \tag{10}$$

where $z$ is the elevation of the ice surface , $\beta_{\mathrm{acc}}$ and $\beta_{\mathrm{abl}}$ are the accumulation and ablation mass balance gradients, respectively, and $\dot{b}_{\mathrm{acc}}^{\max}$ is an upper bound for the accumulation rate.

This parameterization is a simplification of the actual and mostly unknown spatial patterns of accumulation and ablation processes at the LGM. Together with an equation for the surface temperature (next section), we have effectively decoupled the
mass balance from the energy. A rate of ablation based on the number of positive degree-day (PDD model) would have been more physical. However, because of large uncertainties in LGM climate (including annual temperature amplitudes), previous applications of the PDD approach had to rely on present-day temperature distribution minus an offset (e.g. Becker et al., 2016; Seguinot et al., 2016; Jouvet et al., 2017). Furthermore, PDD factors needed to compute surface melt rates are known to vary substantially (cf. Braithwaite, 1995; Hock, 2003; van den Broeke et al., 2010) and choosing suitable factors for LGM conditions
is challenging. The same is true for accumulation which requires knowing patterns of rainfall and temperature. Since LGM climate is known to have behaved rather differently from today owing to the southward displacement of the atmospheric polar front in the North Atlantic (Florineth, 1998; Hofer et al., 2012; Luetscher et al., 2015), it is questionable whether the added complexity of a PDD model results in a more accurate representation of LGM mass balance distribution.

Table 2 lists the chosen mass balance gradients. Different values of $\beta_{\mathrm{acc}}$ and $\dot{b}_{\max}$ were selected to represent a range of
dry to wet climates while values of $\beta_{\mathrm{abl}}$ were chosen to be either in agreement with earlier estimates for the LGM Rhine glacier (Haeberli and Penz, 1985) or with present-day Greenland values (Machguth et al., 2016) measured in regions where temperatures and precipitation are similar to estimated LGM conditions in northern Switzerland.



### 2.5.4 Surface temperature

Because of the uncoupling of mass balance and energy, surface temperature only influences ice temperature and rheology (see Eq. (3)) but not the rate of accumulation or ablation. As done in many ice modelling studies (e.g., Tarasov and Peltier, 1999; Seddik et al., 2012; Seguinot, 2014; Thoma et al., 2014; Jouvet et al., 2017), we assume that the ice surface temperature is equal to the mean annual air temperature. Based on earlier climate reconstruction we assume that the mean annual temperature

at the equilibrium line is $-12°$C, a value within a range of estimations ($-15$ to $-10$ °C, Haeberli, 1983, 2010). We assume that changes in surface temperature with elevation are linearly related to a lapse rate, $\gamma_a$,

$$T_{\text{surf}}(z) = T_{\text{ela}} + \gamma_a \left( z - z_{\text{ela}} \right), \tag{11}$$

where $T_{\text{ela}}$ and $z_{\text{ela}}$ are the temperature and elevation at the equilibrium line, respectively. In all model simulations $\gamma_a = -6°$C km$^{-1}$, an intermediate value based on an estimate by Keller and Krayss (2005b) ($-7°$C km$^{-1}$) for the Rhine glacier

at the LGM and contemporary lapse rates for polar regions ($-5$ to $-4°$C km$^{-1}$, Marshall et al., 2007; Machguth and Cohen, unpublished). Estimates of the equilibrium line altitude at the LGM vary from 944 m (Benz-Meier, 2003) to 1200 m (Haeberli and Penz, 1985). Benz's estimate is based on the assumption that the accumulation area ratio (the ratio of the accumulation area of the glacier to the total surface area) is $2/3$. Another estimate by Keller and Krayss (2005b) using the same method but a slightly different ice surface reconstruction and hypsographic curve found an ELA of 1000 m. In our simulation, ELA ranges

from 1000 to 1200 m (see Table 2).

### 2.6 Lateral boundaries

Nodes on lateral boundaries are either no-flow nodes, where the normal component of velocity perpendicular to the boundary is zero, or outflow nodes, where the natural stress-free boundary condition is applied. Outflow nodes are used on the northern boundary. No-flow nodes are at the glacier basin boundary in the accumulation zone. The horizontal temperature gradient is

assumed to be zero (no flux condition) along all lateral boundaries.

### 2.7 Temperature initialization

One of the most difficult fields to initialize is the englacial temperature (velocities and pressure are easily computed from the Stokes equations once temperature is known). In reality, the temperature of an ice sheet depends on the interplay between climate, flow, and geothermal heating so that the temperature at any instant should require, in theory, knowledge of the full

flow and climate history that affected the ice sheet. For the problem at hand, this is not known. We initialize the temperature field in the ice assuming the temperature depends on an accumulation or ablation rate (vertical advection) and is independent of horizontal advection. The advantage of these assumptions is that an analytical expression for temperature exists from Robin (1955) (see Cuffey and Paterson, 2010, p. 410). The disadvantage of this initialization is that horizontal advection is neglected and thus the temperature distribution is out of equilibrium and simulations necessitate a long spin-up time.





The vertical temperature distribution for an ice thickness between 0 and $H$ with vertical advection given by $w = -\dot{b}z/H$ is, for the accumulation zone ($\dot{b} > 0$),

$$T = T_{\text{surf}} + z_\star \frac{\sqrt{\pi}}{2} \left[\frac{dT}{dz}\right]_{\text{bed}} \left[\text{erf}(z/z_\star) - \text{erf}(H/z_\star)\right], \tag{12}$$

where $[dT/dz]_{\text{bed}}$ is the gradient of temperature at the bed and $z_\star^2 = 2\alpha_T H/\dot{b}$. For ablation ($\dot{b} < 0$), the temperature distribution is

$$\begin{aligned}
\quad T = T_{\text{surf}} + \left[\frac{dT}{dz}\right]_{\text{bed}} &\left\{z_\star \exp(z^2/z_\star^2)\mathcal{D}(z/z_\star) \right. \\
&\left. - z_\star \exp(H^2/z_\star^2)\mathcal{D}(H/z_\star)\right\},
\end{aligned} \tag{13}$$

where $\mathcal{D}()$ is the Dawson integral

$$\mathcal{D}(x) = e^{-x^2} \int_0^x e^{t^2} dt. \tag{14}$$

Using the accumulation/ablation function given in Eq. (10), the ice surface temperature, and a given heat flux, the tempera-
ture distribution in the accumulation and ablation zones is computed everywhere in the glacier using Eqs. (12) and (13). The computed temperature sometimes exceeds the melting temperature when the ice is thick. This is not surprising since nowhere in the model of Robin (1955) there is information about a maximum temperature or about a phase transition. Limiting the temperature to the melting temperature is done as a post-processing step. This obviously breaks apart the conservation of energy, but is a quick fix to obtain a first-order approximation of temperature. Temperatures warmer than the melting temperature are
obtained because, in the accumulation zone, thick ice insulates ice at depth from the cold atmospheric conditions and geothermal heat flux adds heat to the ice from the bottom. In the ablation zone, the process is similar, but in addition, temperate ice is advected upward.

## 2.8   Numerical simulations

Five simulations (Table 2), named s1 through s5, with different values of mass balance gradients (accumulation and ablation)
and ELA, represent a range of climate scenarios from driest (s1) to wettest (s5). Scenario s1 has the smallest accumulation rate and the smallest melt rate at the glacier terminus, representative of an extreme climate, perhaps colder and drier than LGM conditions. Simulation s5, which has the highest cutoff value for the maximum accumulation rate (Eq. (10)), is meant to represent the other extreme: a wetter climate in the south that may have occurred as a result of moisture transport from south of the Alps that yielded significant accumulation in the high Alpine peaks (Luetscher et al., 2015) but that remained
relatively cold and dry in the north near the glacier terminus. Other parameters (basal topography, glaciological parameters) are identical for all simulations. All input parameters and key computed quantities for these simulations are summarized in Table 2. For comparison, the table also includes data for two geomorphic reconstructions: Benz-Meier (2003) and Keller and Krayss (2005b).



**Table 2.** Summary of numerical (this study) and selected geomorphic reconstructions. $z_{\mathrm{ela}}$ is elevation of equilibrium line. $\beta_{\mathrm{abl}}$ and $\beta_{\mathrm{acc}}$ are the ablation and accumulation mass balance gradients, respectively. $\dot{b}_{\mathrm{max}}$ is the maximum accumulation rate. $\bar{b}_{\mathrm{acc}}$ is the average accumulation rate. $\bar{b}_{\mathrm{net}}$ is the glacier net mass balance. $\dot{b}_{\mathrm{term}}$ is the ablation rate at the terminus. AAR is the accumulation area ratio. $A$ is the glacier area, $V$ its volume. $t_s$ is the simulated time. Numerical simulations s1 through s5 are ranked from driest to wettest based on the value of the average accumulation rate, $\bar{b}_{\mathrm{acc}}$. Initial conditions for the ice surface are: the geomorphic reconstruction of Benz-Meier (2003) for s1, a simulation with $\beta_{\mathrm{acc}} = 0.1\ \mathrm{m}\,(100\mathrm{m})^{-1}\,\mathrm{a}^{-1}$, $\beta_{\mathrm{abl}} = 0.2\ \mathrm{m}\,(100\mathrm{m})^{-1}\,\mathrm{a}^{-1}$, and $z_{\mathrm{ELA}} = 1200$ m that ran for 440 years for s2, and a simulation with $\beta_{\mathrm{acc}} = 0.05\ \mathrm{m}\,(100\mathrm{m})^{-1}\,\mathrm{a}^{-1}$, $\beta_{\mathrm{abl}} = 0.2\ \mathrm{m}\,(100\mathrm{m})^{-1}\,\mathrm{a}^{-1}$, and $z_{\mathrm{ELA}} = 900$ m that ran for 907 years for s3, s4, and s5.

| Reconstruction | $z_{\mathrm{ela}}$ | $\beta_{\mathrm{abl}}$ | $\beta_{\mathrm{acc}}$ | $\dot{b}_{\mathrm{max}}$ | $\bar{b}_{\mathrm{acc}}$ | $\bar{b}_{\mathrm{net}}$ | $\dot{b}_{\mathrm{term}}$ | AAR | $A$ | $V$ | $t_s$ |
|---|---|---|---|---|---|---|---|---|---|---|---|
| | m | m (100 m)$^{-1}$ a$^{-1}$ | m a$^{-1}$ | m a$^{-1}$ | m a$^{-1}$ | m a$^{-1}$ | m a$^{-1}$ | % | km$^2$ | km$^3$ | a |
| Numerical | | Inputs | | | | | Outputs | | | | |
| s1 (driest) | 1200 | 0.1 | 0.025 | 0.26 | 0.19 | 0.02 | −0.65 | 69 | 12,389 | 5,551 | 3262 |
| s2 | 1000 | 0.67 | 0.05 | 1.00 | 0.47 | 0.29 | −3.00 | 88 | 15,831 | 10,932 | 2162 |
| s3 | 1200 | 0.3 | 0.05 | 0.90 | 0.49 | 0.19 | −1.95 | 70 | 14,563 | 10,118 | 2204 |
| s4 | 1100 | 0.4 | 0.1 | 1.00 | 0.64 | 0.42 | −2.20 | 84 | 17,706 | 14,300 | 2162 |
| s5 (wettest) | 1200 | 0.2 | 0.1 | 1.80 | 0.82 | 0.51 | −1.30 | 76 | 16,339 | 13,558 | 1540 |
| Geomorphic | | Outputs | | | | | Inputs | | | | |
| BM03[1] | 944 | 0.53 | 0.034 | – | 0.33 | 0* | −2.1 | 67 | 15,990 | 6,516 | – |
| KK05b[2] | 998 | 0.66 | 0.060 | – | 0.50 | 0* | −3.9 | 67 | 16,400 | 6,450 | – |

[1] Benz-Meier (2003). [2] Keller and Krayss (2005b). *Equilibrium is assumed.

Input and output quantities are different for the numerical and the geomorphic reconstructions. For the numerical simulations, model input parameters are: the ELA ($z_{\mathrm{ela}}$), the mass balance gradients ($\beta_{\mathrm{abl}}$ and $\beta_{\mathrm{acc}}$), and the maximum rate of accumulation ($\dot{b}_{\mathrm{max}}$). Model outputs are the average net accumulation ($\bar{b}_{\mathrm{acc}}$), the average net glacier balance ($\bar{b}_{\mathrm{net}}$), the specific balance rate at the terminus ($\dot{b}_{\mathrm{term}}$) estimated at an elevation of 500 m, the accumulation area ratio (AAR, ratio of accumulation area to total area), and the glacier area ($A$) and its volume ($V$). Also indicated in the table is the simulated time, $t_s$, ranging between 1500 and over 3000 years. For the geomorphic reconstructions, inputs and outputs are almost reversed. Inputs are the glacier area and volume obtained from field mapping and inferences of ice surface elevation contours, AAR, $\dot{b}_{\mathrm{term}}$, and $\bar{b}_{\mathrm{net}}$ assumed to be zero (the glacier is in a state of dynamic equilibrium).

For the geomorphic reconstructions, the term $\dot{b}_{\mathrm{term}}$ is calculated from values of summer and mean annual LGM temperatures near the 500 m elevation level indicated in both Benz-Meier (2003) and Keller and Krayss (2005b). These temperatures are converted to melt rates, first by estimating the number of positive degree days (PDD) using the model of Reeh (1991), and then by multiplying the PDD by a PDD factor, here taken equal to $6\ \mathrm{mm}\,\mathrm{PDD}^{-1}$ (Braithwaite, 1995). Both Benz-Meier (2003) and Keller and Krayss (2005b) assume an AAR of 0.67 based on earlier studies of modern Alpine glaciers (e.g., Gross et al., 1977) that assume zero net balance. This assumption, together with the glacier hypsometry, determines the ELA. Using the ELA and



the melt rate at the terminus, one can compute the average ablation gradient, $\beta_{abl}$. Assuming the glacier was at equilibrium, net accumulation equals net ablation. Net ablation can be calculated from the mass balance gradient computed with the method just described and the glacier hypsometry (area distribution of elevation) below the ELA. Using the same procedure but in reverse, the mass balance gradient in the accumulation area, $\beta_{acc}$, can be calculated from the net accumulation (equals the net ablation) and the glacier hypsometry above the ELA. Numbers for the ablation and accumulation mass balance gradients of the geomorphic reconstructions calculated with this method are given in Table 2. Note that the average accumulation rate calculated for Keller and Krayss (2005b) (0.50 m a$^{-1}$, Table 2) is higher than the value cited in Keller and Krayss (2005b) (0.30 m a$^{-1}$) obtained from numbers cited in Haeberli (1991) that assumed LGM precipitation was equal to 20% of today's value (1.5 m a$^{-1}$).

## 2.9 Numerical solutions

The open source software Elmer/ice (Gagliardini et al., 2013) is used to solve the set of equations and their boundary conditions using the finite element method. The general 3-step procedure for initializing and running simulations is as follows:

– Solve the transient Stokes flow equations together with the free surface for 50 years with a constant temperature field given by the initialization. The energy equation is turned off.

– Solve the steady state energy equation only using the velocity field calculated at the end of the 50 years.

– Solve all fields simultaneously in transient mode (flow, energy, free surface) for several thousand years, possibly reaching a steady state. Simulations were stopped due to computational time constraints.

## 3 Model results

### 3.1 Ice extent, surface elevation, and ice thickness

Ice extent and ice surface elevation of simulations s1 through s5 are shown in Fig. 4. In almost all subsequent figures that present numerical simulation results, the geomorphic reconstruction of Benz-Meier (2003) or a numerical simulation based on his ice surface elevation is included for comparison (e.g., Fig. 4a).

Out of the five simulations shown in Fig. 4, simulation s1 appears to have essentially reached steady state (net glacier balance $\dot{b}_{net} = 0.02$ m a$^{-1}$, see Table 2) and simulation s3 seems close to it ($\dot{b}_{net} = 0.19$ m a$^{-1}$). Whether these states are true equilibrium states is debatable as negative feedbacks associated with topography and changes in mass balance with elevation can lead to complete ice sheet collapse in simple models (e.g., Levermann and Winkelmann, 2016). Whether this applies to our model here is unknown. Other simulations remained out of equilibrium conditions during the entire simulation time with the ice mass still increasing. The two simulations that are closest to an apparent steady state (simulations s1 and s3) have an ice surface area smaller than the LGM extent defined by moraines but a margin shape and configuration that resembles the outline of the LGM terminal moraines (e.g., ice-free Hörnli ridge in between the Rhine and Linth lobes, dendritic outline of ice extent).





**Figure 4.** Ice surface elevation in m.a.s.l. for (a) Benz-Meier (2003) and (b–f) simulations s1 through s5 (Table 2). Black contours are every 500 meters. The 2500 m contour is shown in yellow. The magenta contour indicates the equilibrium line altitude. Equilibrium line is 1200 m except for simulations s2 (1000 m) and s4 (1100 m). Topography is shown in ice-free areas with a different color scheme.

Of these two simulations, simulation s3 is closer to the LGM margin with an ice surface area of 14,553 km$^2$ while simulation s1 is only 12,389 km$^2$ (the geomorphic reconstructions yielded values around 16,000 km$^2$). The ice surface area of simulation s5 is nearly identical to the geomorphic reconstruction (16,400 km$^2$). Simulation s4 is slightly larger (17,706 km$^2$); s2 slightly smaller (15,813 km$^2$). Their configurations, however, are significantly different with most Alpine peaks and the Hörnli ridge ice covered, and with an ice margin that is less dendritic. Note that these three simulations (s2, s4, and s5) have not yet reached

5  steady state and have a net positive mass balance (Table 2) with an ice mass that is still growing and a margin that continues to move north, beyond the LGM margin.





Only the modeled ice surface elevation of simulation s1 resembles the geomorphic reconstruction of Benz-Meier (2003)
(Fig. 4a,b). The 2500 m elevation contours (shown in yellow in Fig. 4) of the geomorphic reconstruction and of simulation s1
in the accumulation area are relatively close (within a few kilometers). Also, the 1200 m contour lines (the ELA for simulation
s1) have very similar positions across the piedmont glacier that forms the Linth lobe (the location where the glacier flows
through its narrowest point before spreading onto the lowlands), while for the Rhine lobe the 1200 m contour line bulges out
5   slightly into the foreland in the numerical simulation. The ice surface elevations of simulations s2–s5 are significantly higher
than the geomorphic reconstruction. In simulations s2-s4, the 3000 m elevation contour is found roughly at the same location
as the 2500 m contour in the geomorphic reconstruction. Simulation s5 has a higher ice surface with the 3200 m elevation
contour at roughly the same position as the 2500 m contour of the geomorphic reconstruction. This indicates an ice thickness
500 to 700 meters higher for simulations s2–s5 than in the geomorphic reconstruction or in simulation s1. Ice thickness for all
numerical simulations is shown in Fig. 5. Maximum ice thickness in the geomorphic reconstruction of Benz-Meier (2003) is
slightly greater than 1500 m in the Vorderrhein (see Fig. 3 for location). This same 1500 m contour has only slightly expanded
in simulation s1. However, in all other simulations, maximum ice thickness exceeds 2000 m in the same area.

### 3.2   Ice surface and hypsometric curve

Hypsometric curves are useful to show the area distribution of elevations of the ice surface. Figure 6 shows hypsometric curves
for the 5 numerical simulations and for the Benz-Meier (2003) and Keller and Krayss (2005b) geomorphic reconstructions.
The curves for the numerical simulations are shifted to the right in comparison to those for the geomorphic reconstructions
indicating more ice at higher altitudes. Only simulation s1 converges to the geomorphic reconstructions for high elevations
(above 2500 m). For the numerical simulations, the values of AAR can be read directly from the graph, the AAR being the
intersection of the ELA with the hypsometric curve. All numerical reconstructions have higher AAR than the value of 0.67
assumed for the two geomorphic reconstructions. For simulations s1 and s3 which are close to steady state, their AAR values
are 0.69 and 0.71, respectively. Simulations s2, s4, and s5, which have not reached steady state, have higher AAR values: 0.88,
0.70, and 0.76, respectively (see Table 2). Since these simulations are still in a phase of ice growth, their steady state AAR
values are likely to be smaller. Note also that the lower the ELA, the higher the AAR.

### 3.3   Ice surface speed

Figure 7 shows the computed ice surface velocities. All numerical simulations (s1 through s5, Fig. 7b–f) show patterns of glacial
flow concentrated in the main trunks of Alpine valleys with highest flow velocities in the Rhine glacier and at the confluence of
the Linth and the Walensee glaciers. Flow velocities decrease rapidly as ice fans out into the Rhine lobe but radially-oriented
topographic lows that follow river drainage keep ice moving at higher speeds there. In contrast, calculated surface velocities for
the geomorphically reconstructed ice surface show no such pattern of ice flow (Fig. 7a). Instead, flow is concentrated in zones
of high slope gradients giving rise to unrealistic patterns of ice surface speeds with no apparent channeling of ice through
Alpine valleys. Also, ice is nearly stagnant in a large portion of the Rhine and Linth lobes with surface velocities less than
10 m a$^{-1}$. For that simulation (Fig. 7a), the ice velocity was obtained from the solution of the Stokes flow equations in steady





**Figure 5.** Ice thickness in meters for (a) Benz-Meier (2003) and (b–f) simulations s1–s5 (Table 2).

state with no sliding and with the reconstructed ice surface of Benz-Meier (2003) fixed with the stress-free boundary condition. A uniform ice temperature of $0°\mathrm{C}$ was also used to obtain the same rheological parameter ($A$ in Eq. (3)) as prescribed by Benz-Meier (2003). Although our Stokes flow model is different from the simple one-dimensional model of Benz-Meier (2003), our patterns of ice flow are comparable to his reconstruction (see Benz-Meier, 2003, Fig. 6.10) and ice velocities are comparable in magnitude for the most part (e.g., 2930 and 3130 $\mathrm{m\,a^{-1}}$ at the Sargans diffluence for our and Benz-Meier's reconstructions,

5   respectively), although not for the areas with very high slope gradients near high Alpine peaks where our model predicts velocities in excess of 5000 $\mathrm{m\,a^{-1}}$, far exceeding velocities predicted by Benz-Meier ($\sim 2500$ $\mathrm{m\,a^{-1}}$). Table 3 summarizes maximum speeds for all simulations as well as other dynamical quantities described in the following sections.





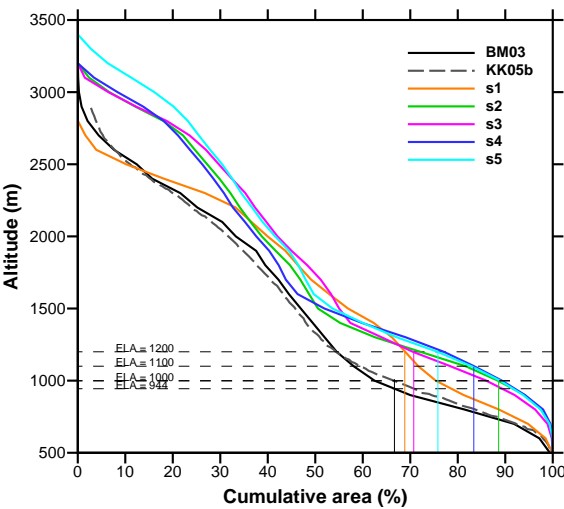

**Figure 6.** Glacier hypsometric curves for geomorphic and numerical reconstructions of the Rhine glacier system. Vertical colored lines show the value of the AAR on the $x$ axis for each reconstructions.

**Table 3.** Summary of ice dynamics quantities. $V^{\mathrm{max}}$ is the maximum surface velocity. $\overline{V}_{\mathrm{lobe}}$ is the average surface velocity in the Rhine lobe. $V_s^{\mathrm{max}}$ is the maximum sliding velocity. $\bar{\tau}_{\mathrm{bed}}$ is the average basal shear stress over the entire glacier bed. VH is the mean vertically integrated strain heating.

| Simulations | $V^{\mathrm{max}}$ | $\overline{V}_{\mathrm{lobe}}$ | $V_s^{\mathrm{max}}$ | $\bar{\tau}_{\mathrm{bed}}$ | VH |
|---|---|---|---|---|---|
| | $[\mathrm{m\,a}^{-1}]$ | $[\mathrm{m\,a}^{-1}]$ | $[\mathrm{m\,a}^{-1}]$ | $[\mathrm{kPa}]$ | $[\mathrm{mW\,m}^{-2}]$ |
| Benz-Meier (2003) | 2930 | $< 1$ | 0 | 73 | – |
| s1 | 248 | 28 | 116 | 91 | 34 |
| s2 | 331 | 57 | 227 | 135 | 98 |
| s3 | 333 | 54 | 284 | 118 | 101 |
| s4 | 357 | 66 | 244 | 161 | 110 |
| s5 | 535 | 79 | 358 | 145 | 193 |

## 3.4 Basal conditions

Basal temperature and sliding speed are two key variables that describe basal conditions. Basal temperature is shown in Fig. 8 as the difference between modelled ice temperature and the pressure melting point calculated as a function of ice pressure at the bed. The small range shown in Fig. 8 ($0.1°\mathrm{C}$) helps to identify zones of temperate basal ice. The yellow contour outlines basal ice that is within $0.05°\mathrm{C}$ of the pressure melting point, essentially temperate for all practical purposes. For the numerical



**Figure 7.** Ice surface speed in m a$^{-1}$ for (a) Benz-Meier (2003) (steady-state Stokes flow solution with no sliding and a uniform ice temperature of 0 °C) and (b–d) simulations s1–s5 (Table 2). Also shown are ice flow lines separating major ice divides. Green: Linth–Walensee; black: Walensee–Rhine; orange: Rhine–Ill. See Sect. 4.3.2 for details and Fig. 1 for location of major glaciers. Note the different scale for speed between (a) and all other subfigures.





**Figure 8.** Basal temperature below the melting point ($dT = T - T\mathrm{pmp}$) in $^\circ$C for (a) Benz-Meier (2003) (temperate bed assumed) and (b–f) simulations s1–s5 (Table 2). Yellow contour indicate the extent of ice that is within $0.05\,^\circ$C of the pressure melting point.

simulation based on the geomorphic ice surface reconstruction (Fig. 8a), the bed is assumed temperate as in Benz-Meier (2003). For all other numerical reconstructions, large areas of the bed of Alpine glaciers and the majority of the Rhine and Linth lobes are at the melting temperature. Exceptions are the upper reaches of Alpine valleys, and areas in the lobes where ice is thin and the cold temperatures at the ice surface diffuse down to the base of the glacier. Note also that, in general, the northeastern part of the Rhine lobe east of the Schussen valley (see Fig. 3 for location) is colder than the western part of the lobe, probably owing to rising hummocky topography in the east (Beckenbach et al., 2014) and to thinner ice cover there (see Fig. 5).

Sliding speed is shown in Fig. 9. Following Benz-Meier (2003), zero basal velocity is assumed for the numerical geomorphic reconstruction (Fig. 9a). For other numerical simulations, the spatial patterns of sliding are directly related to the basal ice temperature (Eqs. 6 and 7). Significant sliding begins in the Rhine valley at the confluence of the Vorderrhein and the Hinterrhein





Figure 9. Ice sliding speed in m a$^{-1}$ for (a) Benz-Meier (2003) (no sliding assumed) and (b–f) s1–s5 (Table 2).

with very limited sliding up-valley of this location. Downstream of the Vorderrhein-Hinterrhein confluence, temperate bed conditions prevail and sliding speeds reach their maximum values in the main trunk of the Rhine valley (Alpenrhein) around Sargans where the Rhine splits into the main Rhine and the Walensee glacier, and also in the lower portion of the Rhine valley at the Alpine gate before ice enters the German-Swiss Forelands and the Lake Constance basin. Sliding is also significant at the confluence of the Linth and Walensee glaciers.

5    Following patterns of surface speed (Fig. 7), sliding speed is highest in the Alpenrhein in simulation s5 (Fig. 9f) reaching about 200 m a$^{-1}$. Maximum sliding speed diminishes for simulations which have smaller ice throughflow and surface slopes. In simulation s1 which has the least amount of ice, maximum sliding speed is less than 100 m a$^{-1}$ at the Sargans diffluence.





In the lobes, sliding speed generally diminishes quickly as ice radially spreads into the lowlands but sliding speed patterns there also follow topographic patterns of troughs and bedrock highs. For example, sliding speed is relatively high ($150 \, \mathrm{m \, a^{-1}}$) in the two main branches (Limmat and Glatt valleys) of the Linth lobe. In the larger Rhine lobe, sliding is significant along the Lake Constance trough and the Thur valley near Frauenfeld (see Fig. 3 for locations), with speeds in excess of $100 \, \mathrm{m \, a^{-1}}$. These patterns are more pronounced for simulation s5 which has the highest ice flux.

Basal shear stress, another indicator of basal condition, is shown in Fig. 10 and is calculated using Eq. (6). For the simulation based on the ice surface reconstruction of Benz-Meier (2003) (Fig. 10a), basal shear stress is extremely low in the Rhine and Linth lobes, not exceeding 0.02 MPa. Outside of the lobes, values in excess of 0.3 MPa are found in areas with high slope gradients and high surface velocity (cf. Fig. 7a). In other numerical simulations (s1–s5), basal shear stress depends on the mass balance. For simulation s1 with the smallest mass balance gradient, computed basal shear stresses is low in the lobe (up to

0.05 MPa) and up to 0.1 MPa in Alpine valleys. Higher values are obtained for other simulations with higher mass balance gradients and higher ice mass turnover. For these simulations, basal shear stress in the lobe can reach up to 0.1 MPa, twice the value computed for s1, and up to 0.3 MPa in Alpine valleys. The large difference in basal shear stress between lobes and Alpine valleys is consistent among all present simulations and is also a feature of earlier models (Haeberli and Penz, 1985). In high elevation areas where ice is frozen to the bed, basal shear stress can also exceed 0.25 MPa. High values of basal shear

stress (up to 0.3 MPa) are in accord with results from other numerical modeling studies (Brædstrup et al., 2016), and with local measurements beneath glaciers (Cohen et al., 2000; Iverson et al., 2003; Cohen et al., 2005). Also the basal shear stress at the margin of the lobes is high owing to a transition from temperate (sliding) to frozen (no slip) basal conditions.

Finally, the ratio of the sliding speed to the surface speed is shown in Fig. 11 as an indicator of internal deformation. This ratio is 0 for the geomorphic reconstruction (Fig. 11a) since no sliding was assumed there. A ratio near 1 indicates little or no

internal deformation. The ratio of sliding to surface speed shows a clear pattern for all simulations. Sliding is proportionally greater (80% of surface speed) in the lobes than in the main trunks of valleys (50–70%). In the lobe, ice flows almost like a plug with little vertical gradients in velocity owing to sliding and to low surface slopes resulting in low driving stresses. The value of the ratio, however, is not homogeneous in the lobe, and mimics patterns of basal temperature with less sliding with respect to surface speed where ice is colder. Sliding ratio increases outward from the center of the lobe up to the cold margin.

Unsurprisingly, thicker ice with a temperate, sliding base has a larger component of ice deformation than thinner ice. In the main Alpine valleys, sliding is about 50 to 60% of surface speed. Slightly higher values are obtained for simulation s1 which has smaller surface gradients and smaller driving forces and thus, less internal ice deformation.

## 4    Discussion

### 4.1    Steady state

Numerical solutions simulated at least 1500 years of glacier evolution and more than 3000 years in one case (see Table 2). Transient Stokes flow simulations are computer intensive and time consuming, practically limiting the number of runs to be performed and the simulation time. Only two simulations achieved near steady-state conditions (simulations s1 and s3). For





**Figure 10.** Basal shear stress in MPa for (a) Benz-Meier (2003) (steady state Stokes solution) and (b–f) simulations s1–s5 (Table 2).

the other three, positive net mass balance (see Table 2) indicates that the ice mass is still increasing and that, at equilibrium, the ice margin would have extended further north than the LGM moraines. This is already the case for simulation s4 (see Fig. 4e).

Although our objective was initially to obtain steady state configurations of the Rhine glacier under different climate scenarios, the actual Rhine glacier may never have reached equilibrium condition at the LGM. The climate record prior to the LGM indicates large oscillations as shown by the oxygen isotopic record ($\delta^{18}$O) in Greenland (Dansgaard et al., 1993; North Greenland Ice Core Project members, 2004) and also from several different proxies in Europe and in the Alps (e.g., Guiot et al., 1993; Genty et al., 2003; Spötl and Mangini, 2006). Two recent speleothems from a cave near Bern, Switzerland, also display these oscillations (Luetscher et al., 2015). These century to millennia timescale oscillations, called Dansgaard-Oscheger (DO) events, lasted through the Marine Isotope Stage 3 (MIS3, 60–30 ka BP) until the LGM (Railsback et al., 2015). The climate



**Figure 11.** Ratio of sliding to surface speed for (a) Benz-Meier (2003) (no sliding assumed) and (b–f) simulations s1–s5 (Table 2).

record around the LGM, although significantly smoother, was not steady. That period lasted around 2000 years in Greenland and in the Alps (see Luetscher et al., 2015, for the Bernese Alps). Probably as a result of the climate variability during the LGM period, the Rhine glacier left two closely-spaced terminal moraines in the Rhine lobe (e.g., Keller and Krayss, 2005a; Ivy-Ochs, 2015; Beckenbach et al., 2014). This two-fold maximum is also observed elsewhere in the Alps, on the Tagliamento glacier in the southern Alps in Italy (Monegato et al., 2007), and in Austria (van Husen, 1997). Formation of these two moraines may

5    have been separated in time by less than a couple of thousand years for the Rhine glacier (see Keller and Krayss, 2005b). Thus, glacier dynamics was likely never in equilibrium with the mass balance. In addition, thermal equilibrium takes significantly more time to establish than dynamic equilibrium. Thus, is it more than likely that the Rhine glacier at the LGM was not a




steady state configuration. This partially validates the use of non-steady solutions as illustrations of possible Rhine glacier configurations at the LGM.

## 4.2 Glacial extent and ice thickness: comparison with geomorphic reconstructions

Two simulations, s2 and s5, have ice cover areas that nearly match the geomorphic reconstruction (16,339 and 15,831 $\text{km}^2$, respectively, against 16,400 $\text{km}^2$), but are out of equilibrium with ice extent still increasing. For these two simulations, however,

the Hörnli ridge is ice covered and the western part of the Rhine lobe is less extensive when compared to the geomorphic reconstruction. The former is due to overall thicker ice for these two simulations. The latter may be due to using the present-day depth of Lake Constance as the basal topography for all of our simulations. Sediments below Lake Constance, however, were partially excavated prior to the LGM as shown by several geologic cross sections based on borehole data (Ellwanger et al., 2011). A deeper basal topography along the Lake Constance trough would have channeled more ice along a southeast to

northwest path, easing ice advance further to the northwest while reducing the flux of ice in the northeastern part of the Rhine lobe.

In addition to glacial extent, glacial volume may help discriminate between numerical results. Except for simulation s1, all other simulations indicate a thicker glacier (by at least 500 meters) than the geomorphic reconstructions. Benz-Meier (2003) and Keller and Krayss (2005b) estimate an ice volume of $\sim$ 6500 $\text{km}^3$, similar to simulation s1 (5500 $\text{km}^3$) but significantly

less than simulations s2–s5 (10,000 to 15,000 $\text{km}^3$, see Table 2). Thicker ice for the Rhine glacier was also obtained by Becker et al. (2016) in a global reconstruction of the ice mass over the European Alps at the LGM, regardless of the precipitation and temperature offsets used. Although simulation s1 yields an ice profile similar to the geomorphic reconstruction (see Fig. 6), there is a discrepancy between simulation s1 mass balance parameters and independent climatic reconstructions for the LGM. For simulation s1, mass balance gradients necessary to obtain such a low profile Rhine glacier ice surface are very small: 0.1

m $(100\,\text{m}^{-1})\,\text{a}^{-1}$ and 0.025 m $100\,\text{m}^{-1}\,\text{a}^{-1}$ for the ablation and accumulation zones, respectively. This value of ablation gradient is identical to the value obtained by Haeberli and Penz (1985) using glacier shear stress and mass balance consideration on the reconstructed LGM glacier surface topography along a flow line that extended from the Vorderrhein to Lake Constance. These mass balance gradients yield an ablation rate at the terminus of 0.65 m $\text{a}^{-1}$ and an average accumulation rate of 0.19 m $\text{a}^{-1}$, numbers representative of an extremely cold and dry climate. Converting these numbers to temperature

and precipitation using PDD (Reeh, 1991) yields a LGM July temperature of about 0 °C and a LGM precipitation between 10 and 20% of today.

These numbers seem to be too small and inconsistent with LGM climate as reconstructed from pollen and other proxy records (e.g., Wu et al., 2007). Both Benz-Meier (2003) and Keller and Krayss (2005b) report summer temperatures of 3.2 and 7°C, respectively, which translates, using the positive degree day model of Reeh (1991) with a PDD factor of 6 mm $\text{PDD}^{-1}$,

into 2.1 and 3.9 m $\text{a}^{-1}$ of melt at the terminus. Using their values of ELA (see Table 2), these numbers yield ablation gradients significantly higher than those of simulation s1: 0.53 and 0.66 m $(100\,\text{m}^{-1})\,\text{a}^{-1}$ for Benz-Meier and Keller and Krayss, respectively, instead of 0.1 m $(100\,\text{m}^{-1})\,\text{a}^{-1}$ for simulation s1. For the accumulation area, using rough estimates of precipitation of Haeberli and Penz (1985) and the area of the accumulation zone based on hypsometric curves yields gradients of





0.034 and 0.06 m $(100 \text{ m}^{-1})$ $\text{a}^{-1}$ for Benz-Meier and Keller and Krayss, respectively, instead of 0.025 m $(100 \text{ m}^{-1})$ $\text{a}^{-1}$ for simulation s1. Gradients calculated on the basis of Benz-Meier (2003) or Keller and Krayss (2005b) suggest warmer and wetter conditions, closer to values used in simulations s2–s5 (see Table 2). These simulations, however, yielded much thicker ice.

The mass balance ablation gradient of simulation s1, simulation that best match the Rhine glacier reconstructed from geomorphic interpretation, is smaller than any present-day values measured in Greenland. Furthermore, the gradients of simulation
s1 are only slightly larger than measured in the McMurdo Dry Valleys of Antarctica. The compilation of Machguth et al. (2016) of surface mass balance gradients in the ablation area of the Greenland Ice Sheet on glaciers and ice caps in Greenland's coastal areas indicate a range between 0.2 m $(100 \text{ m})^{-1}$ $\text{a}^{-1}$ in the very north to 0.45 m $(100 \text{ m})$ $\text{a}^{-1}$ in the central to southwest as well as the very south of Greenland, about 2 to 4.5 times larger than values obtained in simulation s1. Antarctic mass balance ablation gradients of three Antarctic Dry valleys glacier (Fountain et al., 2006; Machguth and Cohen, unpublished) indicate
values between 0.02 and 0.05 m $(100 \text{ m})^{-1}$ $\text{a}^{-1}$, the latter is only one-half of the value used in simulation s1. Estimated melt rates at the terminus of the Greenland Ice sheet and of local glaciers at an altitude similar to the margin of the Rhine glacier lobe range between 1 and 5 m $\text{a}^{-1}$, 1.5 to 7 times larger than the value used in simulation s1. The marginal melt rates in s1 is close to values found in the Antarctic Dry Valleys (Fountain et al., 2006). Extremely low to even inverse mass balance gradients can locally be produced by extreme shadow (glacier margins in deep-cut valleys), heavy debris cover, or strong snow redistri-
bution with deposition of snow at the ice margin. For the Rhine glacier with a large and predominantly clean piedmont lobe (Haeberli and Schlüchter, 1987), however, such special conditions are hardly plausible. Thus, although simulation s1 yields an ice thickness comparable with geomorphic reconstructions, its climate may be too extreme in view of our understanding of climate conditions near the margin of the Rhine glacier at the LGM.

Estimating LGM mass balance gradients and ablation rates based on reconstructed temperature and precipitation patterns
results in values close to the ones used in simulations s2 through s5. These simulations, however, yielded much thicker ice, up to 700 m more, than the geomorphic map of Benz-Meier (2003), Keller and Krayss (2005b), or Bini et al. (2009). Based on this comparison alone, our simulation results indicate that either mass balance gradients were high and the Rhine glacier was thicker or, gradients were small with the implication that the climate was colder and drier than commonly accepted for the LGM. If the former is correct, this would mean that the trimline used to delineate the glacier extent in the accumulation area,
and from which ice surface elevation contour lines are usually drawn (e.g., Benz-Meier, 2003; Kelly et al., 2004; Bini et al., 2009; Wirsig et al., 2016), does not represent the ice surface but an englacial cold-temperate ice transition layer that indicates the limit of glacial erosion (Florineth, 1998). In several, more polar paleo ice sheets (British Isle, Scandinavia, Canada), several studies have pointed out the existence of uneroded bedrock surfaces beneath ice sheets that have survived multiple glaciations under frozen, cold-based basal conditions (e.g., Fabel et al., 2002; Briner et al., 2003; Ballantyne and Stone, 2015). The same
situation is also believed to exist in some places under Antarctica (Creyts et al., 2014). From a geomorphic point of view, whether cold ice was present above the trimline during the LGM remains a subject of debate but has not been entirely ruled out (e.g., Wirsig et al., 2016). Thus, if instead of the numerically computed ice surface, the cold-temperate ice transition that intersects the basal topography is used for comparing the numerical simulations and the geomorphic reconstruction, the fit between the geomorphic ice surface and the englacial transition is significantly different. Figure 12 shows three images





that attempt to compare the geomorphic reconstruction with this cold-temperate ice transition. In Fig. 12a, the ice extent of the geomorphic reconstruction of Benz-Meier (2003) is shown with visible ice-free peaks above the ice surface (in brown). The boundary of these peaks represents the observed cold-temperate transition (the trimline). Figure 12b shows simulation s5 ice cover (here the ice is so thick that almost no peak protrudes above the ice surface) with the outline of the computed cold-temperate transition colored according to its elevation using the same color scale as the ice surface. Finally Fig. 12c shows this same cold-temperate transition superimposed on top of the geomorphic reconstruction (Fig. 12a). The elevation of the cold-temperate transition is significantly lower than the ice surface of simulation s5. In the Alps near the foreland, the transition overlaps with the ice-free peaks of the geomorphic map of Benz-Meier (2003) which emerge from the ice surface (e.g., Alpstein, Churfirsten, see Fig. 12c). The fit between the trimline of the geomorphic reconstruction (Fig. 12a) and the temperate-cold transition of simulation s5 (Fig. 12b) is near perfect. Higher up in the Alps like, for example, along the Glarus Alps, the cold-temperate transition of simulation s5 is at an elevation significantly lower than the trimline of the geomorphic reconstruction (see Fig. 12c) probably because of the accumulation of cold ice at high elevation in simulation s5 that moves englacially. There, the cold-temperate transition underestimates the elevation of the trimline. This discrepancy could easily be explained by the cold temperature at the ice surface used to drive the numerical simulations. Under a warmer and wetter climate that preceded the LGM, thicker temperate ice may have existed in this region with a cold-temperate ice transition located at a higher elevation along the mountain flanks, closer to the measured elevations of the trimlines. Exploring this scenario would require modeling the evolution of the Rhine glacier over longer time periods with a fluctuating climate, something difficult to do with our Stokes flow model. The discrepancy between climate, Rhine glacier ice thickness in the accumulation zone, and trimline reconstruction deserves further attention.

## 4.3 Rhine glacier dynamics

### 4.3.1 Surface speed

Our numerical simulations of the Rhine glacier system at the LGM indicate that the ice mass had a relatively slow mass turnover with maximum surface speeds ranging between 180 m a$^{-1}$ to 535 m a$^{-1}$, values that reflect overall small mass balance gradients. Highest flow speeds are obtained for simulation s5 which has the highest rate of accumulation. These surface speeds are significantly smaller than the maximum surface speed estimated from the geomorphic reconstruction. Differences between the geomorphic (Fig. 7a) and numerical reconstructions (Fig. 7b–f) are due to the evolution of the ice free surface for more than 1000 years in our numerical reconstruction driven by the mass balance forcing (accumulation or ablation) and by ice flow: with time, surface slope gradients become smoother and are glaciologically consistent with the surface mass balance boundary condition giving rise to the expected patterns of complex ice flow along the main glacial valleys and their tributaries, and with highest surface speeds in the main valleys where ice is deepest and at confluences. Maximum surface velocity in simulations s1 to s2 are 3 to 10 times smaller than the results of either Benz-Meier (2003) or our own reconstruction based on his ice surface elevation. The opposite is true for surface velocity in the lobes. Average computed surface velocities for simulations s1–s5 range from 28 to 79 m a$^{-1}$, about 30 to 80 times greater than computed for the geomorphic reconstruction





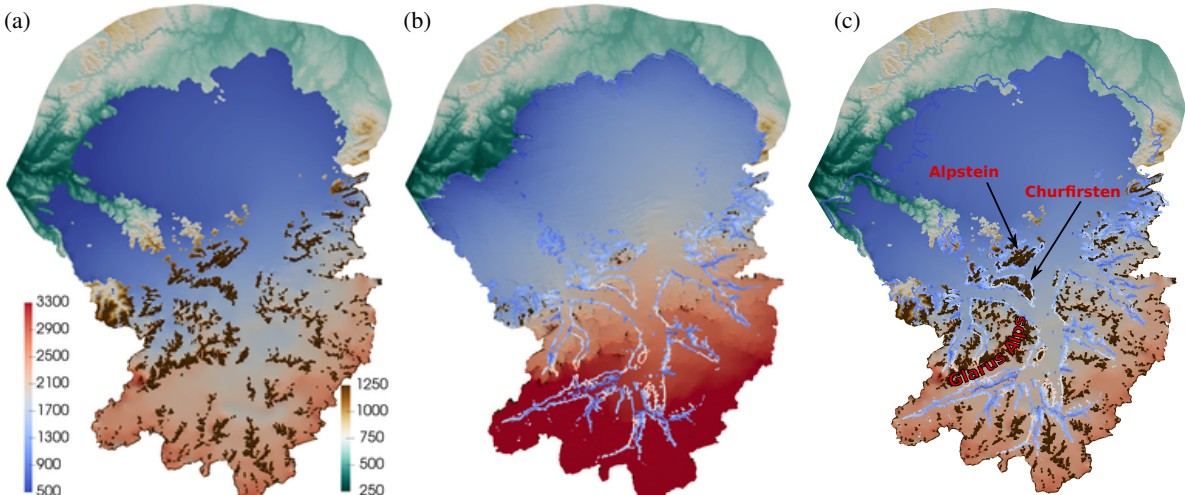

**Figure 12.** Comparison of geomorphic reconstruction with extent of temperate ice for simulation s5. (a) Geomorphic ice surface elevation of Benz-Meier (2003). Brown color represent ice-free peaks above the ice surface; (b) Ice surface elevation of simulation s5 with contour line indicating the position of the temperate-cold transition along mountain flanks and colored according to elevation (same scale as ice surface); (c) Same as (a) with temperate-cold transition of (b) superimposed. All elevations are in meters.

using Stokes flow ($< 1$ m a$^{-1}$, see $\overline{V}_{\text{lobe}}$, Table 3). Such low surface speeds for the geomorphic reconstruction are due to (i) no sliding, and (ii) a nearly flat lobe. An earlier calculation by Haeberli and Penz (1985) that included sliding yielded a value of 25 m a$^{-1}$.

### 4.3.2 Flow patterns and fluxes

Figure 7 shows surface velocity but also included ice flowlines that delineate major ice-forming basins of the Rhine glacier, from the Ill glacier in Austria in the east, to the Linth glacier that fed the western part of the Linth lobe. To obtain streamlines separating basins, streamlines were either computed forward from a confluence of two glaciers (Linth–Walensee, Rhine–Ill) or backward from a diffluence (Rhine–Walensee at Sargans). The streamlines were generated along a vertical line to follow ice motion at different ice depths. Because ice paths vary with depth, streamlines originating at different ice depths sometimes diverge as, for example, the streamlines backtracked from the Sargans diffluence (in black in Fig. 7).

Results indicate that the Rhine lobe is made up of ice originating from the Valserrhein, Hinterrhein, Landquart, and Ill glaciers. Ice from the Vorderrhein moves through the Walensee glacier towards the Linth lobe. This differentiation varies somewhat with the thickness of the ice in the accumulation zone, with some ice from the Vorderrhein making it into the Rhine lobe for the thin-ice simulation (s1) while some ice from the Valserrhein making it to the Linth lobe for the thickest-ice simulation (s5). In the Linth lobe, the Linth–Walensee ice divide sometimes passes west of Pfannenstiel (see Figs. 1 and 3 for locations) through the Limmat valley (simulations s1 and s3), sometimes east of Pfannenstiel through the Glatt valley (simulations s2, s4, and s5). Given the variability, it it possible that this glacier divide was more prone to ice flow



switches depending on climatic and glaciological conditions (e.g., local accumulation rates in the Linth and Rhine basins, basal conditions, etc.). Also of interest, the Schussen lobe, a minor lobe within the greater Rhine lobe (Beckenbach et al., 2014) is made up of ice from the Rhine basin and not from the Ill glacier coming from Austria except for simulation s4 (Fig. 7e) despite its location in the northeastern part of the Rhine lobe.

Additional insight about flow patterns can be gained by generating tracers randomly in a basin and following their paths

downstream. Figure 13 shows ice flowlines colored according to the source region of the ice. Flowlines are calculated from the three-dimensional velocity field. For the geomorphic reconstruction (Fig 13a), the flowlines are based on the velocity field computed after 50 years of transient evolution of the ice surface because the steady state velocity field (see Fig. 7a) does not yield coherent streamlines.

The flowlines computed for Benz-Meier (2003) (geomorphic reconstruction) indicate that ice coming from the eastern

glaciers (Ill, Landquart) dominates in the Rhine lobe (Fig. 13a). In contrast, all numerical simulations (s1–s5) indicate that the Rhine lobe consists mostly of ice from the Hinterrhein and the Vorderrhein (Fig. 13b–f). For the geomorphic reconstruction, ice originating from the Hinterrhein has an almost east to west flow direction in the Rhine lobe. For the numerical simulations (s1–s5) flow of ice from the Hinterrhein has a southeast to northwest orientation. Ice from the Landquart valley is found in the area of Untersee in the geomorphic reconstruction while is it mainly located north towards the Schussen valley in Germany

in numerical simulations. Ice from the Ill valley is found as far west as the Überlingen branch of Lake Constance in the geomorphic reconstruction while it is mainly found further east of Schussen valley in the numerical simulations. In the Linth lobe, ice to the west of the Hörnli ridge originates form the Linth glacier while ice to the east in the Glatt valley originates from the Walensee glacier. In simulations s1 and s3 (lowest mass balance gradients of all simulations), some ice from the Walensee flows to the west of the ridge indicating less ice from the Linth glacier in the eastern part of the model.

Ice thickness also influences flow paths with thicker ice allowing movement of ice across high passes and ridges in the Alps. While flow is constrained by main valleys in simulation s1 with the lowest mass balance gradients and smallest ice thicknesses, the ice surface in other simulations is significantly higher allowing ice to move more directly from the accumulation to the ablation across mountain ranges through high passes, for example across the Alpstein and Churfirsten ranges. Despite these differences, however, the overall patterns of ice flow remain strongly constrained by the general complex topography of the

Alpine valleys.

Three-dimensional velocities across the different basins can help estimate annual fluxes of ice to the Linth and Rhine lobes (Table 4). Fluxes depend on the rate of accumulation. The range of estimated ice flux can vary by a factor of 5 for the Rhine glacier between the simulations with the smallest accumulation rate (s1, 0.93 km³) and the highest rate (s5, 4.8 km³). The Rhine glacier, because of its large basin area, brings most ice to the Rhine lobe (about 80%), with little variation between the

simulations (s1–s5), the remaining ice in the Rhine lobe coming from the Ill glacier. For the Linth lobe, The Walensee glacier brings slightly more ice (0.27 to 1.4 km³, 56–67% depending on the simulation) than the Linth glacier (0.11 to 1.1 km³). Fluxes computed for the geomorphic reconstruction of Benz-Meier (2003) yield values close to the driest simulations (s1) for the Linth lobe and closer to intermediate simulations (s2 and s3) for the Rhine lobe. The origin of the ice in the lobe, however, is different, with equal fluxes of ice coming from the Rhine and the Ill glaciers in the Rhine lobe, and the Linth and the Walensee





**Figure 13.** Three dimensional flow lines for simulations (a) Benz-Meier (2003) after 50 years of free surface evolution and (b–f) s1–s5 (Table 2). Colors indicate origin of ice: (green) Linth glacier; (red) Vorderrhein; (yellow) Hinterrhein; (white) Landquart valley; (orange) Ill glacier. Black lines in (a) indicates the four gates through which ice fluxes are calculated (see Table 4). Background color is elevation of basal surface.

glaciers in the Linth lobe. Given the vastly different basin sizes of these glaciers, these numbers do not appear to be realistic. An estimate of ice flux entering the Rhine lobe by Keller and Krayss (2005b) yielded a value of 1.86 km$^3$, a value higher than the driest simulation (s1, 1.1 km$^3$) but substantially smaller than the next two wetter simulations (s2 and s3, 3.3 and 3.1 km$^3$, respectively).




**Table 4.** Annual ice fluxes for the most important glaciers feeding the Linth and Rhine lobes. Cross-sections (gates) where fluxes are computed are shown in Fig. 13a. For the Benz-Meier (2003) simulation, fluxes are calculated after 50 years of transient simulation with Stokes flow with an evolving ice surface.

| | Linth lobe | | | | Rhine lobe | | | |
|---|---|---|---|---|---|---|---|---|
| Simulation | Linth $km^3$ | Walensee $km^3$ | Total $km^3$ | % Walensee in lobe | Rhine $km^3$ | Ill $km^3$ | Total $km^3$ | % Rhine in lobe |
| Benz-Meier (2003) | 0.26 | 0.25 | 0.51 | 49 | 1.9 | 1.8 | 3.6 | 51 |
| s1 | 0.11 | 0.27 | 0.38 | 71 | 0.93 | 0.21 | 1.1 | 82 |
| s2 | 0.42 | 0.81 | 1.2 | 66 | 2.7 | 0.60 | 3.3 | 82 |
| s3 | 0.40 | 0.82 | 1.2 | 67 | 2.6 | 0.54 | 3.1 | 83 |
| s4 | 0.72 | 0.98 | 1.7 | 58 | 3.2 | 0.91 | 4.1 | 78 |
| s5 | 1.1 | 1.4 | 2.5 | 56 | 4.8 | 1.1 | 5.9 | 81 |

### 4.3.3 Lobe dynamics

As ice from the Rhine glacier enters the Alpine Foreland through the Lake Constance basin, it fans out in a lobe over 60 km long and 100 km wide. In the lobe, ice flows over undulating topography with numerous troughs and valleys separated by topographic highs (see basal topography in Fig. 3). Sliding speed (Fig. 9) is highest in these troughs where ice is thicker (Fig. 5) and basal ice is at the melting point (Fig. 8). In the Rhine and Linth lobes, valley troughs with zones of overdeepenings
focus ice flow while bedrock highs are zones of lower ice speeds.

Because of topography, sliding speed does not decrease monotonically as ice fans out to the margin in the Rhine lobe. Zones of relatively fast sliding occur near the margin owing to topographic effects and also to ice convergence. In simulations with the thickest ice and with the most extensive glacier cover (s2, s4, and s5), ice velocities, particularly sliding speeds, are sometimes higher near the margin than further up glacier. This is the case for the Thur valley near the terminus and for Überlingersee (see
Fig. 9c,e,f and Figs. 1 and 3 for locations). For the Thur valley near the terminus, convergence of ice from Lake Constance and from the upper part of the Thur valley into a low-lying area causes ice to speed up. For the Überlingersee, a narrow trough forces ice to accelerate.

In the Linth lobe the situation is simpler with ice from the Linth and Walensee glaciers flowing between well constrained ridges that separate the Limmat and Glatt valleys. The Albis and Uetliberg mountains delineate the lobe's western flank. In the
center, the Pfannenstiel mountain forces ice to separate and flow around it, and the Hörnli ridge separates the Linth lobe from the Rhine lobe to the east. Except for the value of fluxes and the streamline that separates ice originating from the Linth or Walensee glaciers, flow patterns in the Linth lobe do not change significantly between the different simulations.



**Figure 14.** Extent and thickness in meters (blue to red color map) of temperate basal ice for (a) Benz-Meier (2003) after 50 years of free surface evolution and (b–f) simulations s1–s5 (Table 2). Background green to brown color scheme shows basal topography.

## 4.4   Basal conditions

In accord with earlier work (Blatter and Haeberli, 1984; Haeberli and Penz, 1985), our numerical simulations indicate that the Rhine glacier system is a polythermal glacier. Basal ice is temperate across much of the lobes and in the lower reaches of Alpine valleys and cold elsewhere (see Fig. 8). The temperate basal ice layer, however, is thin, about 100 m in the Alpine valleys and less than 30 m in the lobe (Fig. 14). This represents only a small fraction of the total ice thickness, about 6 to 7 5   % in both valleys and the lobes. Volumetrically, cold ice represents roughly 90% of the total bulk ice but the temperate ice at the bed controls sliding and thus the overall dynamics of the glacier. Contrary to expectations, the temperate basal ice does not grow in thickness down-glacier. The temperate basal ice layer is thickest in the Rhine valley below the Sargans diffluence,





**Figure 15.** Vertically integrated strain heating in W m$^2$ for (a) Benz-Meier (2003) after 50 years of free surface evolution and and (b–f) simulations s1– s5 (Table 2).

probably because of flow constriction associated with the valley width and to higher viscous strain heating there. Increased flow velocity and strain rates cause more heat to be generated englacially bringing cold ice near the bed to the melting temperature as noted in earlier modeling studies (Pohjola and Hedfors, 2003; Lüthi et al., 2015) and in conceptual models (Krabbendam, 2016). This strain heating effect is shown in Fig. 15 which displays the vertically integrated strain heating projected onto the bed. Significant differences in magnitude exist between the thin-ice simulations (simulation with geomorphic ice surface

5  reconstruction and simulation s1, Fig. 15a,b) and all other simulations with thicker ice (simulations s2–s5, Fig. 15c–f) but patterns of high strain heating are similar. The Rhine glacier, from the confluence of the Vorderrhein with the Hinterrhein to Lake Constance, is where strain heating is highest, with values in excess of 1 W m$^{-2}$ for simulations s2–s5 (for comparison, geothermal heat flux is around 0.1 W m$^{-2}$). Flow of ice through Alpine valleys, past constrictions, and around bedrock bumps





increases shear deformation and viscous strain heating. Strain heating is larger at the flanks of Alpine valleys than in the center, reflecting higher shear strain rates there where changes in sliding speeds are greater (see Fig. 9). Also viscous strain heating is large at confluences (e.g., Linth and Walensee, Ill and Rhine), and where ice flows across ridges such as the Appenzell foothills (near where the Rhine glacier enters the Lake Constance basins) or past bedrock bumps like Fläscher Berg near the Sargans diffluence (see Fig. 3 for locations). In contrast, where ice is cold (high Alpine areas) or moves nearly like a plug (Rhine and

Linth lobes), viscous strain heating is negligible.

Zones of high viscous strain heating are also zones of higher basal shear stress (see Fig. 10). This is not surprising since areas with high viscous strain heating are the same areas that undergo high shear strain rate at the glacier base. Valley flanks where basal ice temperature transitions from temperate to cold, bedrock ridges and bumps, confluences, are zones where basal shear stress is highest. Bedrock protuberances act like sticky spots (Alley, 1993) increasing basal resistance. Across these obstacles,

basal shear stress can reach values in excess of $0.25$ MPa (Fig. 10). Where ice is cold and thus sticks to the base, basal shear stresses are also high. In contrast, where it is mostly temperate like in the Rhine and Linth lobes, basal shear stresses are significantly smaller with maximum values between $0.05$ to $0.1$ MPa for thin and thick-ice simulations, respectively.

In one simulation we turned on basal friction (see Eq. (8)). Figure 16 shows the effects of basal friction on basal conditions. Basal friction generates additional heat at the glacier bed (about $0.02$ W m$^{-2}$ on average), increasing basal temperature to the

melting temperature everywhere in the lobe, increasing the sliding speed by a few 10's of m a$^{-1}$ everywhere on the Swiss Plateau. This causes the margin to advance several kilometers further north over the 200 years of the simulation. The effect on the thickness of the temperate ice layer, however, is minimal.

## 5 Conclusions

Using a fully coupled thermodynamics ice flow model (Elmer/Ice), we investigated the dynamics of the Rhine glacier at the

LGM for several climate scenarios characterized by different combinations of ablation and accumulation area mass balance gradients, and compared our simulations of ice thickness and flow (where possible) to geomorphic reconstructions and earlier glaciological studies. Our numerical simulations confirmed results of earlier studies but provided more details of the complex three-dimensional ice flow patterns. In all climate scenarios tested, the Rhine glacier is polythermal with temperate basal ice in Alpine valleys downstream of the Vorderrhein-Hinterrhein junction, and in most of the Rhine and Linth lobes. In all simulations,

cold and dry climatic conditions yielded a LGM Rhine glacier that moved relatively slowly (maximum ice surface speeds of 150 to 530 m a$^{-1}$ with even slower speeds in the lobe (average surface speed of 25 to 80 m a$^{-1}$). Sliding patterns mimicked surface speeds with highest values at the bottom of Alpine valleys. Sliding speed decreased as piedmont glaciers fanned out in the lobes but topographic lows that follow present-day river drainage focused ice flux and yielded complex patterns of ice flow in the lobes. Basal shear stress in the lobe was smaller than $0.1$ MPa for the high mass balance gradient simulations, and

down to $0.025$ MPa when prescribing low mass balance gradients. The latter value is similar to earlier calculations. Higher values, up to $0.3$ MPa were obtained higher up in the glacier and at the margin. In the lobe, a cold margin persisted owing to cold climatic conditions, and ice above several topographic highs was also frozen to the bed. These highs corresponds to old





**Figure 16.** Effect of basal friction on basal temperature, sliding speed, and basal ice thickness for simulation s5 at $t = 1540$ a. (a–c) No basal friction, (d–f) with basal friction. (a,d) Basal temperature. (b,e) Sliding speed. (c,f) Thickness of temperate basal ice layer. Basal friction was turned on for 200 years. Note also the slightly larger extent of ice for the simulation with basal friction on.

uneroded gravel deposits, several hundreds to one million years old, indicating that parts of the bed resisted glacial erosion. Relatively high flow speeds in some areas near the glacier margin are caused by convergence of ice from different parts of the lobe over troughs such as in the area of the Thur valley.

Our results indicate that the geomorphic ice surface in the accumulation area reconstructed from trimlines can only be matched assuming an extremely cold and dry climate, substantially colder and dryer than estimated for the LGM from proxy records. Simulations that used a less extreme cold-dry climate yielded an ice thickness 500 to 700 meters more than the geomorphic reconstruction. This discrepancy can be explained if the trimline and the derived geormorphic ice surface are





interpreted as a cold-temperate ice surface above which glacial erosion did not occur, leaving no erosional features on mountain valley flanks.

Our numerical results indicate that transection glaciers such as the Rhine glacier have complex three-dimensional flow patterns that are difficult to reduce to two dimensions. The coupling of ice flow and thermodynamics yields patterns of basal conditions that are inherently three-dimensional, with patches of cold ice within large areas of temperate basal conditions in the lobes. Early studies of the Rhine glacier captured some of its main features but lacked precision needed to fully understand the coupling between ice flow, climate, and basal thermal conditions. Future models that include the effects of permafrost on basal sliding, and how ice loading and distribution influence patterns of ground water flow are needed to better understand conditions at the LGM and the characteristics of the subsurface during ice-age conditions.

*Acknowledgements.* The project was funded by the Swiss Cooperative for the Disposal of Radioactive Waste (Nagra), Wettingen, Switzerland. Most of the computations presented in this paper were performed using the Froggy platform of the CIMENT infrastructure (https://ciment.ujf-grenoble.fr), which is supported by the Rhône-Alpes region (GRANT CPER07_13 CIRA) and the Equip@Meso project (reference ANR-10-EQPX-29-01) of the program Investissements d'Avenir supervised by the French Agence Nationale pour la Recherche. Thanks are due to Olivier Gagliardini, Thomas Zwinger, and Martina Schäfer for help with Elmer/Ice.



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
