# Peer review of "Numerical reconstructions of the flow and basal conditions of the Rhine glacier, European Central Alps, at the Last Glacial Maximum"

_The Cryosphere, 2017_

## Referee Comment (RC1) · S.J. Marshall (Referee) · 2 Jan 2018

This is a carefully written and beautifully illustrated manuscript that revisits LGM reconstructions of the Rhine Glacier and its catchment region. It is a well-studied glacier and region of Switzerland, which strengthens this study - Cohen et al. build on past LGM reconstructions of the Rhine Glacier, but applying a sophisticated 3D flow model in this case. There is certainly value added in this, with detailed dynamical reconstructions available from the suite of simulations presented in this paper. While the results are mostly of regional interest, the glaciological methods are state-of-the-art and are transferable to other glaciated regions, and there are some robust, general conclusions that

may be of broader interest. In particular, the implications for LGM climate in the area are interesting, and the results also provide a provocative challenge to interpretation of trimline records in alpine regions. If the authors are correct that 100s of metres of inactive, cold-based ice may have been present, with no trace, then icefields in most of the world's mountain regions could have been much thicker than has been assumed or reported elsewhere, on the basis of trimline records.

The climate and mass balance treatments here are not as state-of-the-art as the glaciological modelling, which limits the conclusions about LGM climate. It would be interesting to explore further, e.g. in an RCM, what LGM mass balance gradients and precipitation amounts might have looked like. But that is beyond the current scope. I find this study to be thorough, careful, and well-presented, and most of my comments are included in the attached pdf: highlights for spelling/grammatical errors, and comments/requests for clarification as points arose during my read of the manuscript. I will repeat here a couple of points that I would like to see addressed in the revised text:

1. The question of disequilibrium is difficult in these simulations. I agree with the authors that there is no reason to believe the Rhine glacier complex would have been at equilibrium at LGM or at any time during the glaciation, so the trimlines and moraines rather just represent the maximum thickness (perhaps) and extent. On the other hand, it is hard to interpret the simulations, since they are essentially snapshots along a continuum of glacier/icefield evolution in the region. Numbers in the tables and the thermal and dynamical fields in the various plots are sampling five of an infinite number of potential states, depending on when the simulations were terminated, so what do they mean exactly? It would have been interesting to carry one of the 'most likely' climate/mass balance scenarios out to equilibrium, but I understand the technical constraints. Also, the thicker icefields in runs 3 to 5 may even thicken to where they start to overwhelm the upper topography and challenge the boundary conditions on the upper glacier (i.e. require a larger domain). I would be interested to read a brief discussion of this issue and how the authors interpret their results, perhaps emphasizing that these
are five glaciologically-sensible configurations within a continuous spectrum, but that these do not bound or constrain what is likely or possible.

2. I am a bit uncertain of the sliding treatment and associated discussion. The authors will agree that this treatment, sliding that is linearly proportional to the basal shear stress, is not necessarily the way that large-scale basal flow occurs. For instance, in ice shelves or in water-lubricated environments like ice streams, basal friction and shear stress approach zero as basal flow increases. I appreciate that this is a standard treatment and in the absence of a coupled hydrological model it seems fine, but I would suggest not to over-interpret the basal sliding results. Also, I was confused in places as to the discussion and interpretation on this (p.34, I.15; p.22, I.5); how is basal shear stress gradients in the stress balance? Then sliding is calculated from the resulting value of \tau\_d, per Eq. (6)? (Iteratively). Just a few lines of clarification would help here.

3. p.25, I.27. The climate inferences are interesting. I am not sure about the argument that case 1 is too cold. Values of 4-6 degC are similar to present-day summer temperatures on mid-latitude alpine glaciers, at elevations of 2000-2500 m (e.g., Greuell and Bohm, 1988; Marshall, 2014; Ayala et al., 2015). A temperature of 0 degC does not seem unreasonable for LGM and would represent a cooling of about 5 degC (it is necessary to compare glacier environments rather than the present-day low-elevation temperatures in a non-glacial environment, for the temperature anomaly).

Related to this, the general climate conclusions are of broad interest, I suspect, and I would be interested to read what the authors believe to be most likely for the LGM climate conditions here. How can this be explored further? Is the cold-dry case possible, or can it be ruled out? What is the basis for ruling out a (south)westward source of moisture to the region, from North Atlantic storm tracks displaced to the south relative to present-day, but along the LGM polar front?

TCD
Most of these queries are just looking for additional information and insight from the authors, who have thought about this carefully. Thanks for this interesting contribution.

Please also note the supplement to this comment: https://www.the-cryosphere-discuss.net/tc-2017-204/tc-2017-204-RC1supplement.pdf

---

## Referee Comment (RC2) · Anonymous Referee #2 · 2 Mar 2018

This study simulates the Last Glacial Maximum (LGM) state of the Rhine Glacier using a state-of-the-art numerical ice flow model solving for full Stokes. Five different (simple) climate forcings are applied, and the resulting glacier characteristics are compared to geomorphological reconstructions. The study is interesting; especially the apparent mismatch between the simulated and reconstructed glacier state (simulations using more realistic LGM climate forcing result in a too thick glacier compared to the geomorphological reconstructions). The manuscript is in general clearly written, and very comprehensive. I do have, however, some comments, that I hope the authors will incorporate before the final publication of this work.

GENERAL COMMENTS

My general comments all deal with how the set-up of the model, the initial conditions and the climate forcing might affect the model results. I am basically asking you to explain and discuss this better in the manuscript.

1. Surface mass balance: I understand that the authors prefer to apply a simple surface mass balance (one that is also uncoupled to the surface temperature/energy), because of the uncertainties in simulated and reconstructed LGM surface mass balance and temperature.

(1a) However, to me it is unclear why these particular values for ablation and accumulation gradients, and these equilibrium line altitudes (ELA), are used? The values seem almost randomly chosen, and the five simulations have no logical sequencing of changing one parameter at the time (which would help to better understand the impact).

(1b) Related to Page 12, lines 19-28: This section is slightly confusing. Why is simulation s1 referred to as the "cold" simulation, while it actually has the lowest ELA of all five simulations? The surface temperature is defined by the ELA, or?

(1c) Also the directional component is confusing: "wetter climate in the south" "cold and dry in the north". I thought that the surface mass balance and the surface temperature both only depend on elevation, not on wind or moisture supply direction? Actually, including a directional component might improve the modelled glacier shape to the observations. By imposing a South-North gradient in accumulation, it might become more difficult to glaciate the Hornli ridge (as is now the case in s2, s4 and s5), better fitting the geomorphological observations.

(1d) Page 25, lines 23-26: It me it is unclear how you can calculate temperatures from your surface mass balance, if these are uncoupled. Please explain this more carefully.

2. Initial conditions for ice surface: To me it is not entirely clear which initial ice surfaces are applied. For simulation s1 the reconstruction of Benz-Meier is used, and for

simulations s2-s5, other simulations that ran 440 & 907 years provided the ice surface. In the latter case, are these also based on the Benz-Meier reconstruction? In other words, is the reconstruction always used as basis, followed by 440 or 907 years of simple climate forcing (before simulating s2-s5)? What is the reason for using different initial conditions? I am asking this, because I think that the initial conditions possibly have a strong impact on the model results. But it is difficult to extract this impact due to the (to me random) set-up of the model simulations.

3. Geothermal heat flux: I agree that adjusting and interpreting the geothermal heat flow data available is beyond the scope of this work, but it would be good to see a map of the values used in the simulations. How much does the basal temperature depend on the geothermal heat flux applied? And in how much does this boundary condition of geothermal heat flux define the basal conditions simulated in this study? In other words, does the geothermal heat flux pattern predefine the basal temperature pattern?

4. Steady state: I agree that you should not aim for reaching steady state with your simulations, as indeed climate and ice rarely reach a steady state due to the long response time of the ice compared to climate variability (DO and other variability). It would therefore indeed be unlikely that the Rhine glacier would be in equilibrium with the LGM climate. The argumentation for this (page 23-24) can be written more concisely. Also some studies suggest that DO1 occurred during the last deglaciation, so rather write: ". . . called Dansgaard-Oeschger (DO) events occurred repeatedly during Marine Isotope Stage 3 (MIS3, 60-30 ka BP)." Also, it is difficult to define the duration of the LGM, so I suggest deleting the sentence "That period lasted around 2000 years . . . Bernese Alps)."

SPECIFIC COMMENTS

Page 1, line 3: "fully-coupled"; what do you mean with this? Readers might think that the model is coupled to a climate model – which it is not.

Table 2: Simulated time; why did you not run all simulations the same length of time, or

until they reached the same rates of (dis)equilibrium?

The Abstract starts with mentioning a study about the safety of repositories for radioactive waste, would be nice to come back to that in the conclusions or discussion, and possibly give a recommendation.

TECHNICAL COMMENTS

Figures 1-3 are difficult to compare for none experts of this region. Could you indicate the overlap in the figures, by for example, outline boxes?

Page 10 and Table 2: Please note that the notation of the upper bound for the accumulation rate is not the same.

Fig. 4: would it be possible to indicate the location of the terminal moraines in this figure?

Fig. 4-16: The double colour scale makes some of the figures difficult to understand. I would suggest to discard the ice-free topography, as this is the same in all figures; and make that white. If you do decide to keep the ice-free topography, than please label the colour scales in the figures, and possible use a more dissimilar colour spectrum for the ice-free topographies, as the brown and red are difficult to distinguish.

Fig. 4-16: please delete "(Table 2)" from the caption, not necessary.

Page 15, line 32: "similar" instead of "nearly identical"

Page 22 and fig. 11: Please use either ratio's (0-1) or percentages (0-100%), for consistency.

Fig. 14: This is an interesting figure to compare with Fig. 8. However, it would be clearer if only the extent and thickness of the temperature basal ice was shown, not the basal topography as well.

---

## Author Comment (AC1) · 22 Mar 2018

1. The question of disequilibrium is difficult in these simulations. I agree with the authors that there is no reason to believe the Rhine glacier complex would have been at equilibrium at LGM or at any time during the glaciation, so the trimlines and moraines rather just represent the maximum thickness (perhaps) and extent. On the other hand, it is hard to interpret the simulations, since they are essentially snapshots along a continuum of glacier/icefield evolution in the region. Numbers in the tables and the thermal and dynamical fields in the various plots are sampling five of an infinite number of potential states, depending on when the simulations were terminated, so what do they mean exactly? It would have been interesting to carry one of the most likely climate/mass balance scenarios out to equilibrium, but I understand the technical constraints. Also, the thicker icefields in runs 3 to 5 may even thicken to where they start to overwhelm the upper topography and challenge the boundary conditions on the upper glacier (i.e. require a larger domain). I would be interested to read a brief discussion of this issue and how the authors interpret their results, perhaps emphasizing that these are five glaciologically-sensible configurations within a continuous spectrum, but that these do not bound or constrain what is likely or possible.

We generally agree with the reviewer: our 5 simulations are snapshots along a continuum of transient states of the Rhine glacier around the LGM. Simulation 1 is close to equilibrium (net mass balance of  $0.02 \text{ m a}^{-1}$ ) and the glacier extent and the ice thickness do not change significantly in the last 500 years of the simulation. Simulation 2 is still slightly out of equilibrium but the rate of ice thickness change is small. This is not the case for the other simulations (3 to 5) where the glacier mass balance is still clearly out of equilibrium. For simulations 3 to 5, we chose a configuration of the Rhine glacier whose ice extent was relatively neaer the ice extent obtained from the geomorphic reconstruction. The flow pattern in these simulations is not affected by the lateral boundaries. Had the glacier extent increased further north than in simulation s4, lateral boundaries would have influenced the numerical solutions for the velocity and the ice thickness, and the results would have indeed become unreliable.

Like both reviewers, we believe that the Rhine glacier at the LGM was not at equilibrium with the climate. The five solutions shown are meant to represent five possible maximum extents before the beginning of a climate warming that would have caused the Rhine glacier to retreat from these maximum extents. Our five simulations are part of a continuum of possible state of the Rhine glacier at the LGM. However, these simulations, with a range of ablation and accumulation mass balance gradients, bracket, to some extent, the characteristics of the ice flow for the Rhine glacier at the LGM. Significantly lower or higher mass balance gradients are very unlikely based on what is known about the LGM climate. We do not show or discuss temporal changes in geometry during the simulations. This could fill the work of another paper but we will make clearer in the manuscript that these five simulations are part of a continuum of solutions and illustrate possible states of the Rhine glacier before a climate warming turning point.

 $\mathbf{2}.$ I am a bit uncertain of the sliding treatment and associated discussion. The authors will agree that this treatment, sliding that is linearly proportional to the basal shear stress, is not necessarily the way that large-scale basal flow occurs. For instance, in ice shelves or in water-lubricated environments like ice streams, basal friction and shear stress approach zero as basal flow increases. I appreciate that this is a standard treatment and in the absence of a coupled hydrological model it seems fine, but I would suggest not to overinterpret the basal sliding results. Also, I was confused in places as to the discussion and interpretation on this (p.34, l.15;p.22, l.5; how is basal shear stress calculated in the model? Is it the residual of  $\tau_d$  - lateral drag - longitudinal stress gradients in the stress balance? Then sliding is calculated from the resulting value of  $\tau_d$ , per Eq. (6)? (Iteratively). Just a few lines of clarification would help here.

The reviewer is correct in that our sliding law is a simplification. A linear sliding law does not apply to places like ice streams of where ice separates from bedrock bumps and forms water cavities between ice and rock. Our sliding law, standard in many numerical models of ice flow, uses a linear sliding law as a first order approximation of how a glacier slides on its substrate. Other more complex models of sliding (e.g., Gagliardini et al., 2003) are more realistic but also would have required (to be realistic) some information about subglacial hydrology, which we lacked. Also, these more complex laws, in combination with Stokes flow, are largely untested and we thus preferred keeping the model simpler. A more complex sliding law could be used in follow up studies, keeping in mind that realistic sliding parameterization is

hampered by lack of data against which to verify models. We will check our manuscript to make sure we do not over-interpret results based on sliding. Our results regarding basal temperature, a key parameter that indicates the possibility of glacial erosion, however, appear robust.

The basal shear stress,  $\tau_b$  (shown in Figure 10), is calculated from Equation (6) using the computed velocity field at the bed,  $v_s$ , which is tangential to the bed. The basal drag boundary condition (Equation (6)) adds nonlinearity to the Stokes equation and the velocity solution is indeed calculated iteratively.

On page 34, our use of the term 'basal friction' may be confusing. We meant there the effect of the friction between ice and rock on the thermal boundary condition at the base (Equation (8)) due to the sliding speed. We did not mean the frictional sliding law. We will rephrase this statement to make this clearer.

3. p.25, l.27. The climate inferences are interesting. I am not sure about the argument that case 1 is too cold. Values of 4-6 deg C are similar to present-day summer temperatures on mid-latitude alpine glaciers, at elevations of 2000-2500 m (e.g., Greuell and Bohm, 1988; Marshall, 2014; Ayala et al., 2015). A temperature of 0 deg C does not seem unreasonable for LGM and would represent a cooling of about 5 deg C (it is necessary to compare glacier environments rather than the present-day low-elevation temperatures in a nonglacial environment, for the temperature anomaly).

Our paragraph on page 25 on LGM climate may need some clarifications, but we are not sure we understand the comparison of our LGM summer temperature in the glacier forefield at about 400 m a.s.l. with the presentday summer temperatures at 2000–2500 m in the Alps. The reviewer refers to an elevation range (2000–2500 m) that is roughly 800 m below today's ELA (about 3000 m a.s.l.). Similarly, the LGM glacier forefield (400 m a.s.l) is also roughly 800 m below the LGM ELA (about 1200 m a.s.l.), and this may be the analogy made by the reviewer. In that case, the LGM cooling for our simulation with 0 deg C summer temperature (simulation 1) in the glacier forefield would be between 5 and 10 deg C because present-day temperatures in the Swiss Alps (1961–1990 average) at 2000–2500 m a.s.l. are in the range of 5 to 10 deg C. The cooling could thus be significantly larger than 5 deg C. Also, one should not forget that, even though paleoclimatic reconstruction of summer LGM temperatures at 400 m a.s.l. based on vegetation (in the range of 3–7 deg C) are not much colder than they are today (5–10 deg C) at 2000–2500 m a.s.l. (in both cases 800 m below the ELA), there was less melt at the LGM because the melt season was significantly shorter due to the stronger annual temperature amplitude at the LGM. Finally, and this was our main point, a summer temperature of 0 deg C at the LGM in the foreland is significantly colder and is in conflict with vegetation reconstruction for the same time period. The reconstructed values of 3 to 7 deg C are from the literature and not our own calculations. Assessing the reliability of these values in the context of paleoclimate reconstruction is beyond the scope of this manuscript.

Related to this, the general climate conclusions are of broad interest, I suspect, and I would be interested to read what the authors believe to be most likely for the LGM climate conditions here. How can this be explored further? Is the cold-dry case possible, or can it be ruled out? What is the basis for ruling out a (south) westward source of moisture to the region, from North Atlantic storm tracks displaced to the south relative to present-day, but along the LGM polar front?

The primary goal of our numerical simulations was to reconstruct past ice conditions rather than past climate. Our assumed climate are meant to roughly cover possible climate conditions (specifically mass balance), as input to driving the ice dynamics model, not to make statements about which climate is most likely. Yet, the conclusions presented in our paper indicate a dichotomy between model results and geomorphic reconstructions that is very likely due to climate (specifically mass balance), and that we could not resolve. We are unable to choose a most-likely climate scenario and a cold-case scenario cannot be entirely ruled out at this stage. Further modeling studies, perhaps with different spatial parameterization of mass balance gradients and different sliding laws, may help bring models and geomorphic reconstructions into better agreement. A south-west source of moisture was also not ruled out in our analysis. Indeed more moisture arrived from the south-west at the LGM than today. Under present-day conditions, humidity source is mainly from the northwest. We will check our manuscript to make sure these important points are made.

Also, climate proxies at the LGM, which are rare, may help better bracket the mean annual temperature, winter and summer temperatures, and humidity, at the LGM. Finally, dating of exposure of rocks above the reconstructed trimline should help elucidate whether the trimline was an englacial feature and bring light on the thickness of the Rhine glacier (and other glaciers in the Alps) at the LGM.

**Other comments in annotated manuscript**

**Too high value for PDD factor**

We use a PDD factor of 6 mm day-1 C-1. This is a normal value for melting ice on Arctic glaciers. PDD factors can be a bit higher on mountain glaciers (between 6 to 8 mm day-1 C-1), but we believe that the climate at the LGM was more similar to present-day Arctic conditions. Furthermore, using a higher PDD factor increases the mass balance gradient and makes it more difficult to model a glacier that is in agreement with glacier reconstructions based on geomorphological evidence. We will better indicate in the manuscript the reason for our particular choice of PDD factor.

**Binary map for basal temperature in Figure 8**

Indeed this is like a binary map and we will modify the figure to make that clearer.

**Non-intuitive: increase in friction would produce increased slip**

We agree that increasing friction should decrease slip but in the sliding law we are using, sliding speed increases monotonically with basal shear stress. An alternative explanation, that could also apply to more realistic sliding laws where friction plateaus with increased sliding, is that, with more frictional heating due to higher basal shear stress, a larger area of the bed reaches the melting point, removing potential cold sticky spots and reducing basal resistance, allowing the ice to slide more rapidly.

**Reference**

Gagliardini, O., Cohen, D., Raback, P., and Zwinger, T.: Finite- element modeling of subglacial cavities and related friction law, J. Geophys. Res., 112, F02027, doi:10.1029/2006JF000576, 2007.

---

## Author Comment (AC2) · 22 Mar 2018

**(1a) However, to me it is unclear why these particular values for ablation and accumulation gradients, and these equilibrium line altitudes (ELA), are used? The values seem almost randomly chosen, and the five simulations have no logical sequencing of changing one parameter at the time (which would help to better understand the impact).**

Values of ELA and accumulation and ablation gradients were not chosen at random. A starting point was the estimated LGM ELA altitude of 1200 m by Haeberli and Penz (1985) and Haeberli and Schlüchter (1987), and the value of about 1000 m by both Benz-Meier (2003) and Keller and Krayss (2005). Thus we choose a range of ELA between 1000 and 1200 m. The realistic range of accumulation/ablation gradient together with the ELA value that yields a reasonable Rhine glacier is actually relatively small and our values bracket that range. Given the computation time (weeks) necessary for a Stokes flow simulation at such a high spatial resolution, we only show 5 simulations that reached over 1500 years. Other simulations with shorter simulation times were started but not completed, for the most part because the parameters rapidly yielded too large and unrealistic ice extents. We think that, given the difficulty in obtaining these simulations (time constraint), our sampling of five is more than is usually obtained for this type of ice flow simulation. A systematic investigation of parameter space is unfeasible for such a large computing problem, at least today.

**(1b) Related to Page 12, lines 19-28: This section is slightly confusing. Why is simulation s1 referred to as the cold simulation, while it actually has the lowest ELA of all five simulations? The surface temperature is defined by the ELA, or?**

We think the reviewer meant that s1 had the highest ELA (1200 m). Simulation s1 has the smallest surface melt rate at the terminus owing to the smallest ablation mass balance gradient. This necessarily arises from colder conditions in comparison to other simulations with higher ablation gradients. The surface temperature is indeed defined by the ELA (and thus colder with decreasing ELA), but because we have decoupled the surface temperature from the mass balance, and mass balance effects on glacier flow surpass those of temperature, we rank this simulation as the coldest one based on melt rate at the terminus alone. All simulations are labeled cold or warm based on the mass balance values, not the surface temperature. We will state this point in our revision.

**(1c) Also the directional component is confusing: wetter climate in the south cold and dry in the north. I thought that the surface mass balance and the surface temperature both only depend on elevation, not on wind or moisture supply direction? Actually, including a directional component might improve the modeled glacier shape to the observations. By imposing a South-North gradient in accumulation, it might become more difficult to glaciate the Hornli ridge (as is now the case in s2, s4 and s5), better fitting the geomorphological observations.**

Indeed our surface mass balance and temperature only depend on elevation and not on direction. However, since north-south gradients in accumulation are not known precisely, and since elevation generally increases southward in the model, our high accumulation mass balance gradient simulation could be the result of a combined increased in mass balance with elevation and with southern orientation. Our comment was meant to be conceptual and not quantitative.

**(1d) Page 25, lines 23-26: It is unclear how you can calculate temperatures from your surface mass balance, if these are uncoupled. Please explain this more carefully.**

The mass balance and temperature are uncoupled in the numerical model for the purpose of computing the ice flow (we believe this is ok because the effect of temperature is of a lower order than the mass balance). However, we can still compute melt rates (and using a PDD factor recover summer temperatures) from the imposed mass balance gradients and mass balance at the terminus. This is what is done in that section. We will try to make that clearer.

**(2) Initial conditions for ice surface: To me it is not entirely clear which initial ice surfaces are applied. For simulation s1 the reconstruction of Benz-Meier is used, and for simulations s2-s5, other simulations that ran 440 and 907 years provided the ice surface. In the latter case, are these also based on the Benz-Meier reconstruction? In other words, is the reconstruction always used as basis, followed by 440 or 907 years of simple climate forcing (before simulating s2-s5)? What is the reason for using different initial conditions? I am asking this, because I think that the initial conditions possibly have a strong impact on the model results. But it**

is difficult to extract this impact due to the (to me random) set-up of the model simulations.

The reviewer is correct. For the simulations indicated to use a previous simulation, these previous simulations always started from the Benz-Meier (2003) ice thickness and extent. We will make that clearer in our revision.

**(3). Geothermal heat flux: I agree that adjusting and interpreting the geothermal heat flow data available is beyond the scope of this work, but it would be good to see a map of the values used in the simulations. How much does the basal temperature depend on the geothermal heat flux applied? And in how much does this boundary condition of geothermal heat flux define the basal conditions simulated in this study? In other words, does the geothermal heat flux pattern predefine the basal temperature pattern?**

Although the geothermal heat flux varies between 60 and 120 mW/m$^2$ over our model area of the Rhine glacier, the high values of geothermal heat flux are not, to first order, correlated with temperate basal conditions. Ice flow and climate are the first-order controls. Temperate basal conditions are found upvalley in the Alps where the geothermal heat flux is significantly lower than in the Swiss lowlands occupied by the Rhine lobe. We could include the map of geothermal heat flux we used to argue this point but that would add an extra figure. Instead we propose to add a few explanation in the text.

**(4) Steady state: I agree that you should not aim for reaching steady state with your simulations, as indeed climate and ice rarely reach a steady state due to the long response time of the ice compared to climate variability (DO and other variability). It would therefore indeed be unlikely that the Rhine glacier would be in equilibrium with the LGM climate. The argumentation for this (page 23-24) can be written more concisely. Also some studies suggest that DO1 occurred during the last deglaciation, so rather write: . . . called Dansgaard-Oeschger (DO) events occurred repeatedly during Marine Isotope Stage 3 (MIS3, 60-30 ka BP). Also, it is difficult to define the duration of the LGM, so I suggest deleting the sentence That period lasted around 2000 years . . . Bernese Alps.**

Points well taken. We will modify the text accordingly.

**Specific comment**

**Page 1, line 3: fully-coupled; what do you mean with this? Readers might think that the model is coupled to a climate model which it is not.** Here, and generally in glaciological modelling, the term is meant to indicate the coupling between ice flow and ice temperature (thermo-mechanically coupled) because sometimes ice temperature is assumed (e.g. constant). We will make that point clearer.

**Simulated time; why did you not run all simulations the same length of time, or until they reached the same rates of (dis)equilibrium?**
The main reason is due to the long computational time necessary for the simulations. Each simulation takes several weeks of computer time in a parallel processing environment. Also, in simulations 3 to 5, ice extent increased past the LGM margin, the solution became unreliable because of lateral boundary effects, and the simulation was terminated.

**TECHNICAL COMMENTS**

**Figures 1-3 are difficult to compare for non experts of this region. Could you indicate the overlap in the figures, by for example, outline boxes?** Point well taken. We will add the outline of the Rhine glacier on Figures 1–3.

**Page 10 and Table 2: Please note that the notation of the upper bound for the accumulation rate is not the same.** Correct. Will be changed.

**Fig. 4: would it be possible to indicate the location of the terminal moraines in this figure?** Figure 4a, which reproduces the geomorphic reconstruction of Benz-Meier (2003), follows the terminal moraine. We will make this point clear in the figure caption.

**Fig. 4-16: The double color scale makes some of the figures difficult to understand. I would suggest to discard the ice-free topography, as this is the same in all figures; and make that white. If you do decide to keep the ice-free topography, than please label the colour scales in the figures, and possible use a more dissimilar colour spectrum for the ice-free topographies, as the brown and**

**red are difficult to distinguish.**

We don't find the two color schemes confusing except perhaps in Fig 14 (reviewer's last point below) but we will take this comment into consideration for the final version of the manuscript.

**Fig. 4-16: please delete (Table 2) from the caption, not necessary.**

We will delete these words.

**Page 15, line 32: similar instead of nearly identical**

We will modify the text.

**Page 22 and fig. 11: Please use either ratios (0-1) or percentages (0-100), for consistency.**

We will change the text.

**Fig. 14: This is an interesting figure to compare with Fig. 8. However, it would be clearer if only the extent and thickness of the temperature basal ice was shown, not the basal topography as well.**

Point well taken. We will modify the figure accordingly.

**References**

Benz-Meier, C.: Der würmeiszeitliche Rheingletscher – Maximalstand. Digitale Rekonstruktion, Modellierung und Analyse mit einem Geographischen Informationssystem, Ph.D. thesis, Universität Zürich, 2003.

Haeberli, W. and Penz, U.: An attempt to reconstruct glaciological and climatological characteristics of 18 ka BP Ice Age glaciers in and around the Swiss Alps, Zeitschrift für Gletscherkunde und Glacialgeologie, 21, 351–361, 1985.

Haeberli, W. and Schlüchter, C. (1987): Geological evidence to constrain modelling of the Late Pleistocene Rhonegletscher, Swiss Alps. The Physical Basis of Ice Sheet Modelling. Proceedings of the Vancouver Symposium, August 1987. IAHS, 170, 333–346.

Keller, O. and Krayss, E.: Der Rhein-Linth-Gletscher im letzten Hochglazial. 2. Teil: Datierung und Modelle der Rhein-Linth-Vergletscherung. Klima-Rekonstruktionen, Vierteljahrsschrift der Naturforschenden Gesellschaft in Zürich, 150, 69–85, 2005.

---

## Author Response (AR1)

**Responses to reviewers**

**Reviewer 1**

**1. The question of disequilibrium is difficult in these simulations. I agree with the authors that there is no reason to believe the Rhine glacier complex would have been at equilibrium at LGM or at any time during the glaciation, so the trimlines and moraines rather just represent the maximum thickness (perhaps) and extent. On the other hand, it is hard to interpret the simulations, since they are essentially snapshots along a continuum of glacier/icefield evolution in the region. Numbers in the tables and the thermal and dynamical fields in the various plots are sampling five of an infinite number of potential states, depending on when the simulations were terminated, so what do they mean exactly? It would have been interesting to carry one of the most likely climate/mass balance scenarios out to equilibrium, but I understand the technical constraints. Also, the thicker icefields in runs 3 to 5 may even thicken to where they start to overwhelm the upper topography and challenge the boundary conditions on the upper glacier (i.e. require a larger domain). I would be interested to read a brief discussion of this issue and how the authors interpret their results, perhaps emphasizing that these are five glaciologically-sensible configurations within a continuous spectrum, but that these do not bound or constrain what is likely or possible.**

We generally agree with the reviewer: our 5 simulations are snapshots along a continuum of transient states of the Rhine glacier around the LGM. Simulation 1 is close to equilibrium (net mass balance of 0.02 m a$^{-1}$) and the glacier extent and the ice thickness do not change significantly in the last 500 years of the simulation. Simulation 2 is still slightly out of equilibrium but the rate of ice thickness change is small. This is not the case for the other simulations (3 to 5) where the glacier mass balance is still clearly out of equilibrium. For simulations 3 to 5, we chose a configuration of the Rhine glacier whose ice extent was relatively near the ice extent obtained from the geomorphic reconstruction. The flow pattern in these simulations is not affected by the lateral boundaries. Had the glacier extent increased further north than in simulation s4, lateral boundaries would have influenced the numerical solutions for the velocity and the ice thickness, and the results

would have indeed become unreliable.

Like both reviewers, we believe that the Rhine glacier at the LGM was not at equilibrium with the climate. The five solutions shown are meant to represent five possible maximum extents before the beginning of a climate warming that would have caused the Rhine glacier to retreat from these maximum extents. Our five simulations are part of a continuum of possible state of the Rhine glacier at the LGM. However, these simulations, with a range of ablation and accumulation mass balance gradients, bracket, to some extent, the characteristics of the ice flow for the Rhine glacier at the LGM. Significantly lower or higher mass balance gradients are very unlikely based on what is known about the LGM climate. We do not show or discuss temporal changes in geometry during the simulations. This could fill the work of another paper.

We have added text to indicate that our five simulations attempt to bracket possible configurations of the Rhine glacier at the LGM, (see page 25, lines 26–33), and that the final configurations shown in our study can be interpreted as possible configurations before the termination of the LGM (page 25, lines 8–10).

**2. I am a bit uncertain of the sliding treatment and associated discussion. The authors will agree that this treatment, sliding that is linearly proportional to the basal shear stress, is not necessarily the way that large-scale basal flow occurs. For instance, in ice shelves or in water-lubricated environments like ice streams, basal friction and shear stress approach zero as basal flow increases. I appreciate that this is a standard treatment and in the absence of a coupled hydrological model it seems fine, but I would suggest not to overinterpret the basal sliding results. Also, I was confused in places as to the discussion and interpretation on this (p.34, l.15; p.22, l.5); how is basal shear stress calculated in the model? Is it the residual of $\tau_d$ - lateral drag - longitudinal stress gradients in the stress balance? Then sliding is calculated from the resulting value of $\tau_d$, per Eq. (6)? (Iteratively). Just a few lines of clarification would help here.**

The reviewer is correct in that our sliding law is a simplification. A linear sliding law does not apply to places like ice streams of where ice separates from bedrock bumps and forms water cavities between ice and rock. Our sliding law, standard in many numerical models of ice flow, uses a linear sliding law as a first order approximation of how a glacier slides on its substrate. Other more complex models of sliding are more realistic but also would have required (to be realistic) some information about subglacial hydrology, which we lacked. Also, these more complex laws, in combination with Stokes flow, are largely untested and we thus preferred keeping the model simpler. A more complex sliding law could be used in follow up studies, keeping in mind that realistic sliding parameterization is hampered by lack of data against which to verify models. Our results regarding basal temperature, a key parameter that indicates the possibility of glacial erosion, however, appear robust.

We have added text to indicate that our linear model does not always describe properly basal ice motion (relationship between sliding speed and basal shear stress) but that, given the uncertainties in sliding rules, linear sliding is still appropriate for paleo ice flow models (see text added page 9, lines 6–11). Statements were also added in several places in the manuscript to remind the reader that our sliding and basal shear stress results were obtained for linear sliding.

The basal shear stress, $\tau_b$ (shown in Figure 10), is calculated from Equation (6) using the computed velocity field at the bed, $v_s$, which is tangential to the bed. The basal drag boundary condition (Equation (6)) adds non-linearity to the Stokes equation and the velocity solution is indeed calculated iteratively.

Our use of the term 'basal friction' was confusing. We meant there the effect of the friction between ice and rock on the thermal boundary condition at the base (Equation (8)) due to the sliding speed. We did not mean the frictional sliding law. We rephrased this statement to make this clearer, see page 35 line 17–18.

**3. p.25, l.27. The climate inferences are interesting. I am not sure about the argument that case 1 is too cold. Values of 4-6 deg C are similar to present-day summer temperatures on mid-latitude alpine glaciers, at elevations of 2000-2500 m (e.g., Greuell and Bohm, 1988; Marshall, 2014; Ayala et al., 2015). A temperature of 0 deg C does not seem unreasonable for LGM and would represent a cooling of about 5 deg C (it is necessary to compare glacier environments rather than the present-day low-elevation temperatures in a non-glacial environment, for the temperature anomaly).**

We are not sure we understand the comparison of our LGM summer temperature in the glacier forefield at about 400 m a.s.l. with the presentday summer temperatures at 2000–2500 m in the Alps. The reviewer refers to an elevation range (2000–2500 m) that is roughly 800 m below today's ELA (about 3000 m a.s.l.). Similarly, the LGM glacier forefield (400 m a.s.l) is also roughly 800 m below the LGM ELA (about 1200 m a.s.l.), and this may be the analogy made by the reviewer. In that case, the LGM cooling for our simulation with 0 deg C summer temperature (simulation 1) in the glacier forefield would be between 5 and 10 deg C because present-day temperatures in the Swiss Alps (1961–1990 average) at 2000–2500 m a.s.l. are in the range of 5 to 10 deg C. The cooling could thus be significantly larger than 5 deg C. Also, one should not forget that, even though paleoclimatic reconstruction of summer LGM temperatures at 400 m a.s.l. based on vegetation (in the range of 3–7 deg C) are not much colder than they are today (5–10 deg C) at 2000–2500 m a.s.l. (in both cases 800 m below the ELA), there was less melt at the LGM because the melt season was significantly shorter due to the stronger annual temperature amplitude at the LGM. Finally, and this was our main point, a summer temperature of 0 deg C at the LGM in the foreland is significantly colder and is in conflict with vegetation reconstruction for the same time period. The reconstructed values of 3 to 7 deg C are from the literature and not our own calculations. Assessing the reliability of these values in the context of paleoclimate reconstruction is beyond the scope of this manuscript.

We have modified the end of the section to better explain how temperature and melt rate at the terminus of the glacier are calculated: see see page 26, lines 23–29.

**Related to this, the general climate conclusions are of broad interest, I suspect, and I would be interested to read what the authors believe to be most likely for the LGM climate conditions here. How can this be explored further? Is the cold-dry case possible, or can it be ruled out? What is the basis for ruling out a (south) westward source of moisture to the region, from North Atlantic storm tracks displaced to the south relative to present-day, but along the LGM polar front?**

The primary goal of our numerical simulations was to reconstruct past ice conditions rather than past climate. Our assumed climate are meant to roughly cover possible climate conditions (specifically mass balance), as input to driving the ice dynamics model, not to make statements about which climate is most likely. Yet, the conclusions presented in our paper indicate

a dichotomy between model results and geomorphic reconstructions that is very likely due to climate (specifically mass balance), and that we could not resolve. We are unable to choose a most-likely climate scenario and a cold-case scenario cannot be entirely ruled out at this stage. Further modeling studies, perhaps with different spatial parameterization of mass balance gradients and different sliding laws, may help bring models and geomorphic reconstructions into better agreement. A south-west source of moisture was also not ruled out in our analysis. Indeed more moisture arrived from the south-west at the LGM than today. Under present-day conditions, humidity source is mainly from the northwest. A sentence with extra reference was added on page 3, lines 10–12.

Also, climate proxies at the LGM, which are rare, may help better bracket the mean annual temperature, winter and summer temperatures, and humidity, at the LGM. Finally, dating of exposure of rocks above the reconstructed trimline should help elucidate whether the trimline was an englacial feature and bring light on the thickness of the Rhine glacier (and other glaciers in the Alps) at the LGM. Some comments have been added in the conclusion (see conclusion page 37).

**Other comments in annotated manuscript**

**Too high value for PDD factor**
Text and references have been added to explain our choice of PDD, see page 14, line 14–17.

**Binary map for basal temperature in Figure 8**
We have modified the color scheme for the basal temperature to show a wider range (now from −20 to 0 Celsius, temperature below the melting point). The basal ice that is temperate is outline by a yellow contour. See Figure 8 page 21.

**Non-intuitive: increase in friction would produce increased slip**
We have modified the text to clarify that it is not increased friction that causes the increase in the sliding speed but the heat generated by the basal friction that brings more basal ice in the lobe to the melting temperature, removing cold sticky spots and allowing ice to slide more rapidly. See new text on page 35, lines 18–20.

**Reviewer 2**

**(1a) However, to me it is unclear why these particular values for ablation and accumulation gradients, and these equilibrium line altitudes (ELA), are used? The values seem almost randomly chosen, and the five simulations have no logical sequencing of changing one parameter at the time (which would help to better understand the impact).**

Values of ELA and mass balance gradients were not chosen at random. A paragraph explaining our choices of mass balance gradient and LEA was already included in the original manuscript (now located at the end of page 10 and on page 11) New text was added to explain our choice and also why no parametric study could be performed. Long computation times make it impossible to study the effect of a parameter (keeping all others constant) on the numerical solution. Instead we chose 5 simulations that sample a wide but realistic range of values of mass balance gradients and ELA. We believe that, given the difficulty in obtaining these simulations (time constraint), our sampling of five simulations is more than is usually obtained for this type of paleo ice-flow model. A systematic investigation of parameter space is unfeasible for such a large computing problem, at least today. See text added on page 12 lines 23–28.

**(1b) Related to Page 12, lines 19-28: This section is slightly confusing. Why is simulation s1 referred to as the cold simulation, while it actually has the lowest ELA of all five simulations? The surface temperature is defined by the ELA, or?**

Text has been added to clarify this point. Our notion of cold/warm and dry/wet is based entirely on the mass balance values and not on the ice surface temperature. This is valid since surface temperature is decoupled from the mass balance in our model. See text added on page 13, lines 7–9.

**(1c) Also the directional component is confusing: wetter climate in the south cold and dry in the north. I thought that the surface mass balance and the surface temperature both only depend on elevation, not on wind or moisture supply direction? Actually, including a directional component might improve the modeled glacier shape to the observations. By imposing a South-North gradient in accumulation, it might become more difficult to glaciate the Hornli**

ridge (as is now the case in s2, s4 and s5), better fitting the geomorphological observations.

Text was added to indicate that, although our surface mass balance model only depends on elevation and not on a north-south gradient, since elevation generally increases southward in the model, our high accumulation mass balance gradient simulation could be the result of a combined increased in mass balance with elevation and with southern orientation. We make clear that this is is conceptual idea. See text added on page 13 lines 4–7.

**(1d) Page 25, lines 23-26: It is unclear how you can calculate temperatures from your surface mass balance, if these are uncoupled. Please explain this more carefully.**

Text was added to explain our calculations. See page 26 lines 22–29.

**(2) Initial conditions for ice surface: To me it is not entirely clear which initial ice surfaces are applied. For simulation s1 the reconstruction of Benz-Meier is used, and for simulations s2-s5, other simulations that ran 440 and 907 years provided the ice surface. In the latter case, are these also based on the Benz-Meier reconstruction? In other words, is the reconstruction always used as basis, followed by 440 or 907 years of simple climate forcing (before simulating s2-s5)? What is the reason for using different initial conditions? I am asking this, because I think that the initial conditions possibly have a strong impact on the model results. But it is difficult to extract this impact due to the (to me random) set-up of the model simulations.**

Text was added in the figure caption of Table 2 to indicate the initial conditions of the simulations and in the main text to state that these different initial conditions have little impact on the overall conclusions of our study. See Table 2 and page 13/14 lines 11–12/1.

**(3). Geothermal heat flux: I agree that adjusting and interpreting the geothermal heat flow data available is beyond the scope of this work, but it would be good to see a map of the values used in the simulations. How much does the basal temperature depend on the geothermal heat flux applied? And in how much does this boundary condition of geothermal heat flux define the basal conditions simulated in this study? In other words, does the geothermal heat**

**flux pattern predefine the basal temperature pattern?**

Although the geothermal heat flux varies between 60 and 120 mW/m$^2$ over our model area of the Rhine glacier, the high values of geothermal heat flux are not, to first order, correlated with temperate basal conditions. Ice flow and climate are the first-order controls. Temperate basal conditions are found upvalley in the Alps where the geothermal heat flux is significantly lower than in the Swiss lowlands occupied by the Rhine lobe. Text was added to explain these results on page 34/35 lines 9–10/1–2.

**(4) Steady state: I agree that you should not aim for reaching steady state with your simulations, as indeed climate and ice rarely reach a steady state due to the long response time of the ice compared to climate variability (DO and other variability). It would therefore indeed be unlikely that the Rhine glacier would be in equilibrium with the LGM climate. The argumentation for this (page 23-24) can be written more concisely. Also some studies suggest that DO1 occurred during the last deglaciation, so rather write: . . . called Dansgaard-Oeschger (DO) events occurred repeatedly during Marine Isotope Stage 3 (MIS3, 60-30 ka BP). Also, it is difficult to define the duration of the LGM, so I suggest deleting the sentence That period lasted around 2000 years . . . Bernese Alps.**

We have only modified the text as suggested by the reviewer. See page 25 line 17. No other change was made to the original text.

**Specific comment**

**Page 1, line 3: fully-coupled; what do you mean with this? Readers might think that the model is coupled to a climate model which it is not.** Changed to "thermo-dynamically coupled". See abstract, page 1, line 3.

**Simulated time; why did you not run all simulations the same length of time, or until they reached the same rates of (dis)equilibrium?**

The main reason is due to the long computational time necessary for the simulations. Each simulation takes several weeks of computer time in a parallel processing environment. Also, in simulations 3 to 5, ice extent increased past the LGM margin, the solution became unreliable because of

lateral boundary effects, and the simulation was terminated. A sentence regarding simulation time has been added on page 12 lines 25–28.

**TECHNICAL COMMENTS**

**Figures 1-3 are difficult to compare for non experts of this region. Could you indicate the overlap in the figures, by for example, outline boxes?**

We have added boxes in Figure 1 that outline the locations of Figures 2 and 3.

**Page 10 and Table 2: Please note that the notation of the upper bound for the accumulation rate is not the same.**

Corrected. See Table 2.

**Fig. 4: would it be possible to indicate the location of the terminal moraines in this figure?**

Figure 4a, which reproduces the geomorphic reconstruction of Benz-Meier (2003), follows the terminal moraine. Text added to the figure to make that point clear. See changes in caption of Figure 4.

**Fig. 4-16: The double color scale makes some of the figures difficult to understand. I would suggest to discard the ice-free topography, as this is the same in all figures; and make that white. If you do decide to keep the ice-free topography, than please label the colour scales in the figures, and possible use a more dissimilar colour spectrum for the ice-free topographies, as the brown and red are difficult to distinguish.**

Color scheme for the land surface elevation was changed to gray scale and labelled in all figures.

**Fig. 4-16: please delete (Table 2) from the caption, not necessary.**

Done.

**Page 15, line 32: similar instead of nearly identical**

Done. See page 15 line 18.

**Page 22 and fig. 11: Please use either ratios (0-1) or percentages**

**(0-100), for consistency.**

Changed to ratios everywhere.

**Fig. 14: This is an interesting figure to compare with Fig. 8. However, it would be clearer if only the extent and thickness of the temperature basal ice was shown, not the basal topography as well.**

We have changed the basal topography to a gray scale in all figures which makes the extent and thickness of basal ice easier to see.

---

## Author Response (AR2)

**Responses to editor**

Figure 2 Outline: changed to plural

**Page 13, Line 4** Indeed using a sloping plane for  $Z_{ela}$ , as done by Adalgeirsdóttir et al (2003) would have been a possible way to create a north-south gradient in precipitation. Today in the Alps, ELA depends on distance from the main weather divide (e.g., Huss et al., 2015) with lower ELA closer to the divide. This could probably be quantified using available data for the climate of the last 50–100 years of so. How to do that at the LGM is unclear; different weather pattern may have produced a different distribution of ELA. Clearly there was a stronger north-south gradient than today. Until higher resolution regional climate simulations for the LGM yield the precipitation and temperature distribution at a sufficiently fine resolution to estimate the distribution of elevation of the ELA, this may prove to be a difficult task. However, we keep this idea in mind for the future when such simulation become available and added a sentence in the manuscript to cite the work of Adalgeirsdóttir et al.

Page 15, Line 11. Weertman 1961 reference added.

Page 18, Line 5. Word change. Done.